# Matrix stiffness controls lymphatic vessel formation through regulation of a GATA2-dependent transcriptional program

Maike Frye[1], Andrea Taddei[2], Cathrin Dierkes[3], Ines Martinez-Corral [1], Matthew Fielden[4], Henrik Ortsäter[1], Jan Kazenwadel[5], Dinis P. Calado[2], Pia Ostergaard [6], Marjo Salminen[7], Liqun He[8], Natasha L. Harvey[5], Friedemann Kiefer[3,9] & Taija Mäkinen[1]

Tissue and vessel wall stiffening alters endothelial cell properties and contributes to vascular dysfunction. However, whether extracellular matrix (ECM) stiffness impacts vascular development is not known. Here we show that matrix stiffness controls lymphatic vascular morphogenesis. Atomic force microscopy measurements in mouse embryos reveal that venous lymphatic endothelial cell (LEC) progenitors experience a decrease in substrate stiffness upon migration out of the cardinal vein, which induces a GATA2-dependent transcriptional program required to form the first lymphatic vessels. Transcriptome analysis shows that LECs grown on a soft matrix exhibit increased GATA2 expression and a GATA2-dependent upregulation of genes involved in cell migration and lymphangiogenesis, including VEGFR3. Analyses of mouse models demonstrate a cell-autonomous function of GATA2 in regulating LEC responsiveness to VEGF-C and in controlling LEC migration and sprouting in vivo. Our study thus uncovers a mechanism by which ECM stiffness dictates the migratory behavior of LECs during early lymphatic development.

[1] Department of Immunology, Genetics and Pathology, Uppsala University, Dag Hammarskjölds väg 20, 751 85 Uppsala, Sweden. [2] Immunity and Cancer Laboratory, The Francis Crick Institute, 1 Midland Road, NW11AT London, UK. [3] Max Planck Institute for Molecular Biomedicine, 48149 Münster, Germany. [4] Department of Applied Physics, KTH Royal Institute of Technology, Albanova University Center, 106 91 Stockholm, Sweden. [5] Centre for Cancer Biology, University of South Australia and SA Pathology, SA5000 Adelaide, South Australia, Australia. [6] Lymphovascular Research Unit, Molecular and Clinical Sciences Institute, St George's University of London, SW170RE London, UK. [7] Department of Veterinary Biosciences, University of Helsinki, 00014 Helsinki, Finland. [8] Department of Neurosurgery, Tianjin Neurological Institute, Key Laboratory of Post-Neuroinjury Neuro-Repair and Regeneration in Central Nervous System, Ministry of Education and Tianjin City, Tianjin Medical University General Hospital, Tianjin 300052, China. [9] European Institute for Molecular Imaging (EIMI), University of Münster, Waldeyerstr. 15, 48149 Münster, Germany. Correspondence and requests for materials should be addressed to T.Mäk. (email: taija.makinen@igp.uu.se)

Cells are exposed to different types of mechanical forces such as shear, stretch and matrix stiffness that synergize with chemical cues to regulate cell behavior and fate during development and homeostasis[1]. Cells recognize and respond to these physical stimuli through their cell–cell and cell–matrix adhesions and translate the mechanical information into biological responses in a process called mechanotransduction. For example, extracellular matrix (ECM) stiffness regulates the differentiation of multipotent mesenchymal stem cells. Rigid matrices mimicking bone were found to be osteogenic while soft matrices mimicking brain were neurogenic[2]. Substrate stiffness is also a critical determinant of the ability of stem cells to self-renew[3]. Besides influencing cell fate and behavior at the single cell level, ECM stiffness can regulate tissue morphogenesis[4,5]. This is exemplified by soft matrix driven spatial organization of germ layers during gastrulation[5]. Physical properties of tissues often change in disease. ECM stiffness has been consequently shown to contribute to various diseases including tissue fibrosis, as well as cancer progression by changing cancer and stromal cell functions[6,7].

Endothelial cells (ECs) comprise the inner layer of blood and lymphatic vessels. ECs are surrounded by an extracellular basement membrane (BM) that provides physical and chemical guidance cues for the formation and stabilization of vessel networks[8]. Together with the interstitial matrix (IM), which comprises the interstitial space between all cell types, the BM forms the ECM. The composition and mechanical properties of the ECM differ across the vascular tree, in its surrounding tissues and at different stages of development. The role of specific ECM molecules in vascular development has been studied, and tissue and vessel wall stiffening has been shown to alter endothelial behavior and contribute to vascular dysfunction in disease[9]. However, it is not known if and how ECM stiffness influences vascular morphogenesis. In vitro studies demonstrate that soft matrices induce profound changes in EC shape and behavior by promoting cell elongation, sprouting and capillary network formation, independently of exogenous growth factors[10,11]. Like most adherent cells, ECs respond to soft matrix by reduced proliferation[12]. Key regulators of cell responses to mechanical cues are the YAP and TAZ transcription factors that localize to nucleus and activate targets upon mechanical stimulus, such as stiff ECM, stretching or shear. YAP and TAZ promote cell proliferation in most cell types, including ECs[13–15]. Interestingly, a specific role for TAZ was identified in lymphatic endothelial cells (LECs) in controlling their response to oscillatory shear stress (OSS), which provides a stimulus for the initiation of luminal valve formation[16]. OSS induces LEC quiescence through FOXC2 induction, and loss of FOXC2 leads to TAZ-dependent cell cycle entry and defective valve morphogenesis[14,16]. Although both fluid shear stress and stiffening of the ECM activate mechanosignalling in the EC, it is not known to what extent the cellular responses to the two stimuli are shared.

Here we uncover a novel mechanism by which matrix stiffness controls the critical early step of lymphatic vascular morphogenesis when LEC progenitors delaminate from the cardinal vein and migrate to the surrounding tissue to form the first lymphatic vessels. We identify the GATA2 transcription factor as a critical regulator of matrix stiffness induced transcriptional program in the LECs. As opposed to the previously reported activation of GATA2 by increased mechanical stimulus upon exposure of BECs or LECs to stiff matrix or oscillatory flow, respectively[17–19], we found that GATA2 expression is increased in LECs grown on a soft matrix. We further show that GATA2 is required for early lymphatic vascular morphogenesis by controlling lymphangiogenic growth factor responsiveness following the exposure of migrating venous-derived LECs to a soft embryonic tissue.

## Results

**Venous LEC progenitors experience soft matrix outside the CV.** The first lymphatic vessels form through transdifferentiation of venous to lymphatic ECs followed by their migration out of certain major veins such as the cardinal vein (CV)[20,21]. Venous ECs including the venous LEC progenitors expressing the PROX1 transcription factor are tightly attached to the underlying basement membrane and show a flattened spread out morphology. In contrast, upon exiting the CV LEC progenitors form a network of cells that exhibit a spindle-like shape and elongated nucleus[20] (Fig. 1a). Rigid matrices have been shown to support cell–substrate attachment and cell spreading, while soft matrices promote cell–cell connectivity and endothelial network formation[10]. We therefore hypothesized that the observed change in cell shape and organization reflects exposure of migrating LECs to a softer matrix.

To visualize the mechanical properties of the ECM in vivo, we analyzed deposition of collagen I, which is the predominant collagen type of the ECM and contributes to determining tissue stiffness in the embryo[4,22]. Analysis of immunostained transverse vibratome sections of E11 embryos showed an overall low level of collagen I (Fig. 1b). Arteries and blood capillaries exhibited the highest collagen I density, while CV and the migrating LECs showed moderate or low collagen I, respectively (Fig. 1b, c). ECM stiffness regulates cell shape but also the morphology and structure of the nucleus[23]. On soft matrix, nuclear lamina is distorted and wrinkled, whereas on stiff matrix nuclear distortions are smoothened out[24]. Staining of E11 embryos for LaminB, the major component of the nuclear lamina[25], showed higher nuclear circularity (i.e. lower distortion) in ECs of the CV compared to cells in the surrounding avascular tissue (Fig. 1d, e).

For a more precise measurement of tissue stiffness, we performed ex vivo atomic force microscopy (AFM). The position of the CV and area of the future lymphatic plexus were located using a fluorescence microscope in freshly prepared 350 μm sections of E11 *Prox1-GFP*+ embryos based on GFP fluorescence in the CV as well as the spinal cord and the dorsal root ganglia. The stiffness of the tissue was determined via AFM-based indentation measurement with a pyramidal tip, allowing the values of the effective Young's modulus to be calculated. The CV showed a 13-fold higher stiffness with a Young's modulus of 3.6 ± 0.04 kPa compared to 0.27 ± 0.01 kPa in the surrounding tissue (Fig. 1f). The latter values match those measured in early chicken embryos (0.3 kPa)[4] and adult brain (0.33 kPa)[26], but are significantly lower compared to values from most other adult tissues including muscle (8–17 kPa) or bone (25–40 kPa)[2], or tissue culture plastic or glass (gigapascal range)[27].

Collectively, these results demonstrate that venous LEC progenitors experience a dramatic 13-fold decrease in matrix stiffness during a critical early step in development when they delaminate from the CV and migrate to the surrounding tissue to form the first lymphatic structures. ECM mechanosensing may thus play an important role in regulating LEC phenotype and behavior during early lymphatic development.

**The transcription factor GATA2 is upregulated on soft matrix.** To identify regulators of LEC response to matrix stiffness, we analyzed global transcriptional changes in primary human dermal LECs seeded on stiff (25 kPa) or soft (0.2 kPa) matrices. Affymetrix GeneChip analysis revealed regulation of 2771 transcripts above or below a 1.4-fold change (log2 fold change > 0.5 or < −0.5) threshold on soft versus stiff matrices. We focused on the 162 regulated transcription factors (TFs) of which the potential core TFs controlling LEC response to matrix stiffness were predicted based on up- and downregulated genes with the online

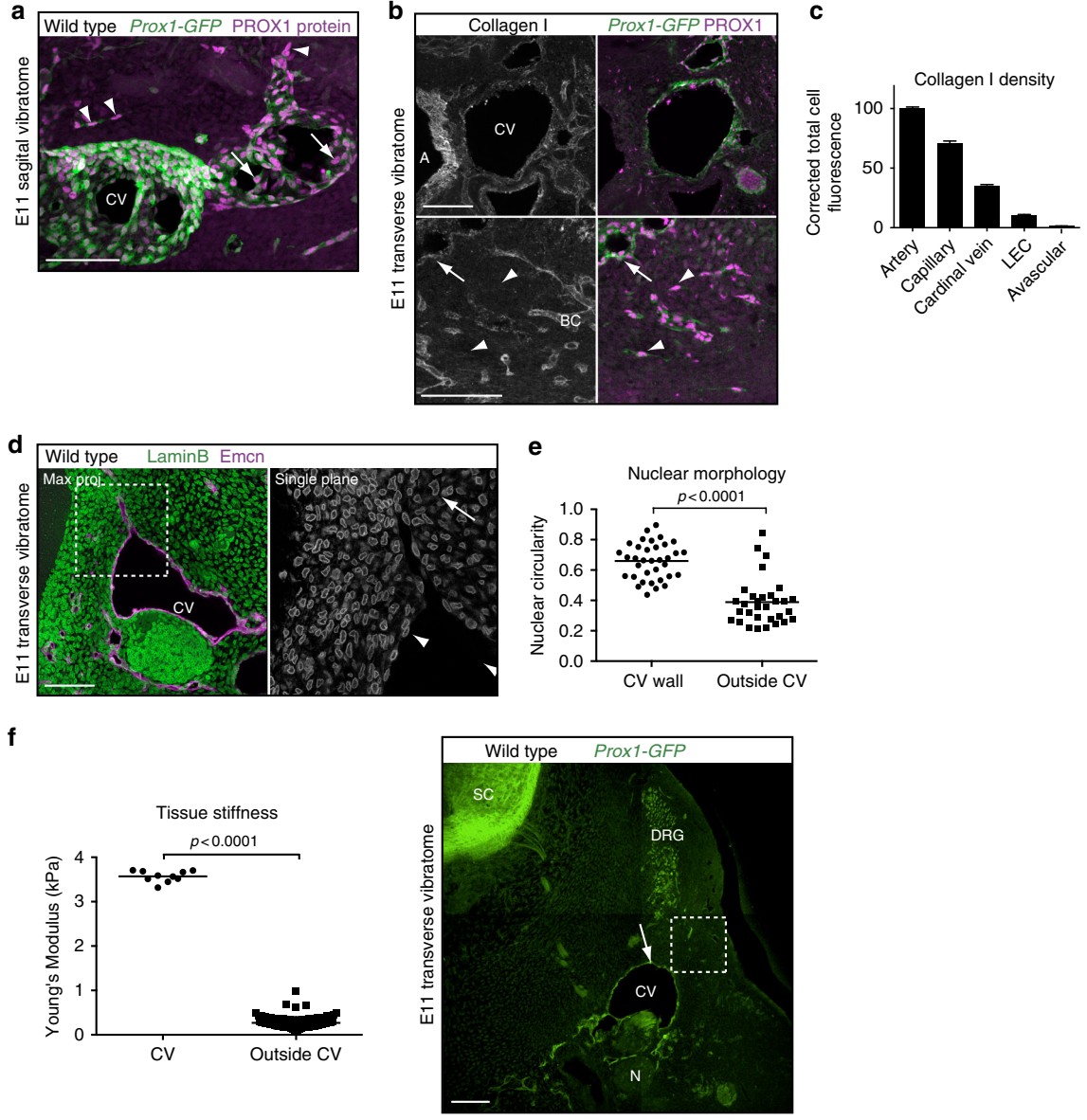

**Fig. 1** Venous LEC progenitors experience lower matrix stiffness upon migration out of the cardinal vein. **a**, **b** Immunofluorescence of sagittal (**a**) and transverse (**b**) vibratome sections of E11 *Prox1-GFP* embryos using antibodies against PROX1 (magenta), GFP (green, *Prox1* reporter) and Collagen I (single channel images in B only). In **a**, note the round nuclear shape of PROX1+ LECs in the cardinal vein (CV) (arrows) compared to elongated nuclei of the migrating PROX1+ LECs (arrowheads) in the surrounding tissue. In **b**, aorta (A) and blood capillaries (BC) show higher Collagen I levels compared to CV (arrow) and migrating LECs (arrowheads). **c** Quantification of Collagen I density in the respective vessel types and avascular tissue in E11 embryos. Data represent mean integrated density values of corrected total cell fluorescence ± s.e.m. (unpaired Student's *t*-test) quantified from $n = 10$ images taken from two embryos. **d** Immunofluorescence of transverse vibratome sections of E11 wild type embryos using antibodies against Emcn (magenta; marker of venous EC) and LaminB (green, marker of nuclear envelope). Single plane image for LaminB staining is shown as a close-up (grey). Nuclear morphology is distorted in cells outside the CV (arrow), as opposed to a round morphology in cells of the CV vessel wall (arrowhead). **e** Quantification of nuclear circularity of Emcn+ venous ECs (CV wall) and Emcn− cells of the surrounding tissue (outside CV). Horizontal lines represent mean ($n = 30$ (CV wall) and $n = 34$ (outside CV) from two embryos). *p* value, unpaired Student's *t*-test. **f** Ex vivo AFM measurements (graph on the left) in transverse vibratome sections (image on the right) of E11 *Prox1-GFP* embryos. Young's Modulus (kPa) is a measure for the actual tissue stiffness. Horizontal lines represent mean ($n = 10$ measurements from one embryo (CV) and $n = 114$ measurements from three embryos (outside CV)). *p* value, unpaired Student's *t*-test. Measurements were done on the dorsal side of the CV (arrow) and the area of LEC migration (boxed area, outside CV). *Prox1-GFP*+ spinal cord (SC), dorsal root ganglion (DRG) and nerve (N) were used for orientation. Scale bars: 100 μm

resource TFactS[28] (Supplementary Data 1). These included the mechanosensitive GATA2 transcription factor that was previously shown to regulate the LEC response to shear stress[18,19], and respond to ECM-derived mechanical signals in blood ECs (BECs) by changing its nuclear/cytoplasmic localization[17]. Furthermore, GATA2 has been identified as a causative gene for Emberger syndrome, a form of primary lymphedema with myelodysplasia (OMIM #614038)[29,30], and a critical regulator of lymphatic valve formation[18].

Quantitative real time (RT)-PCR analysis of human (Fig. 2a) and mouse (Fig. 2b) primary dermal LECs seeded on soft (0.2 kPa) in comparison to stiff (25 kPa) matrices validated the array data and showed a 2-fold increase in *GATA2* mRNA levels. Similar results were observed when analyzing human LECs

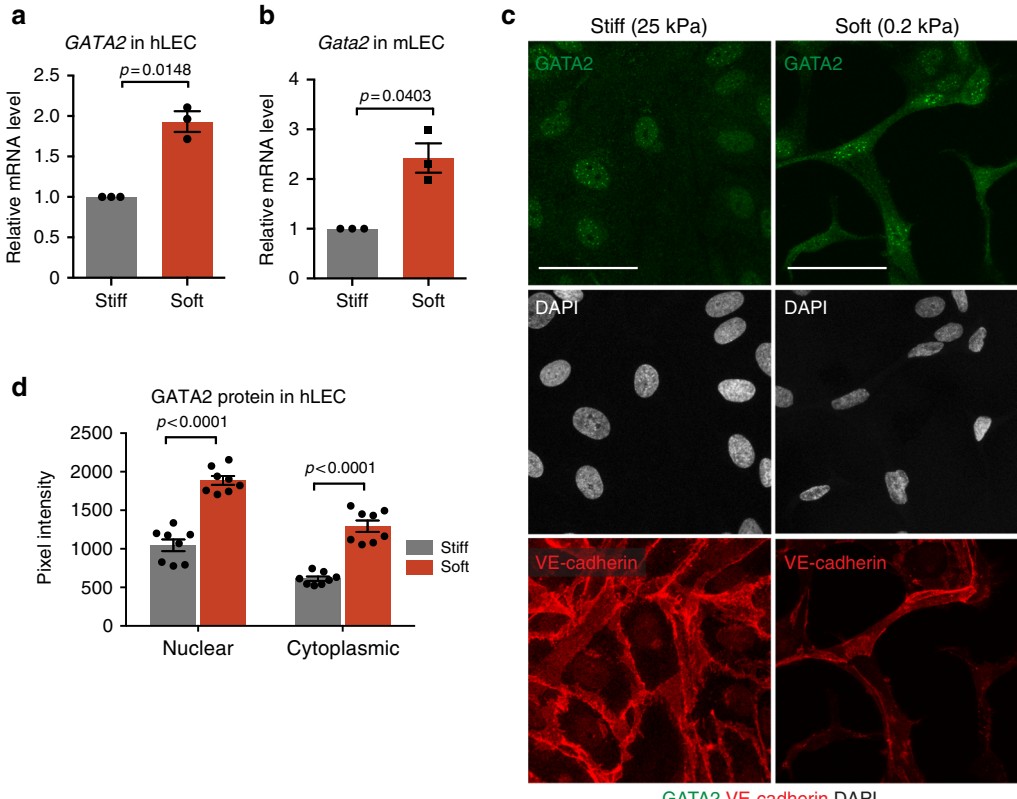

**Fig. 2** GATA2 regulation by matrix stiffness in LECs. **a**, **b** qRT-PCR analysis of GATA2 in human LECs (**a**) and primary mouse LECs (**b**) grown on soft (0.2 kPa) or stiff (25 kPa) matrix. $n = 3$ experiments, mean ± s.e.m. $p$ value, one-sample $t$-test. **c** Immunofluorescence of human LECs grown on stiff and soft matrix using antibodies against GATA2 (green) and VE-cadherin (red), and for DAPI to show nuclei (grey). LECs grown on soft matrix exhibit an overall higher expression of GATA2 as indicated by higher immunofluorescence intensity in both the nucleus and cytoplasm and have an elongated shape and a distorted nucleus. **d** Quantification of nuclear and cytoplasmic GATA2 protein in human LECs grown on soft (0.2 kPa) or stiff (25 kPa) matrix. Data represent mean pixel intensity ($n = 8$ images with 8–24 cells per image (soft), and $n = 8$ images with 21–37 cells per image (stiff) from 3 experiments) ± s. e.m. $p$ value, unpaired Student's $t$-test. Scale bars: 50 μm

seeded on matrices of 0.2 kPa in comparison to those of kPa stiffness representing the physiological stiffness of the embryonic CV (Supplementary Fig. 1a). In agreement with observations in the BECs[17], GATA2 showed a predominantly nuclear localization in LECs seeded on matrices stiffer than 4 kPa (Fig. 2c, Supplementary Fig. 1b). However, unlike in the BECs, GATA2 was not excluded from the nucleus of LECs grown on soft matrices (Fig. 2c). Instead, GATA2 protein levels were increased approximately 2-fold in both the nucleus (1.8-fold) and the cytoplasm (2.1-fold) of LECs grown on soft matrix (Fig. 2c, d).

**GATA2 is upregulated in LECs that migrate out of the CV**. To investigate if GATA2 is functionally important during migration of LEC progenitors, we first analyzed its expression in vivo in ECs of the CV and forming lymphatic vessels. To this end, we dissected jugular regions of E11 embryos carrying the *Prox1-GFP* transgene that labels ECs within the CV, including LEC progenitors, as well as LECs outside of the CV (Fig. 1a). The latter were further identified by the expression of PDPN[20]. mRNA expression in sorted ECs was normalized to the pan-endothelial marker *Tek*, which was not regulated in LECs by matrix stiffness (Supplementary Fig. 1c). As expected, *Prox1-GFP*+PDPN+ ECs showed a strong increase in the expression of the LEC markers *Vegfr3* and *Pdpn* when compared to the expression in *Prox1-GFP*+PDPN− ECs (Fig. 3a). *Gata2* levels were also increased in the PDPN+ LECs by 2.5-fold (Fig. 3a). Expression of the pan-endothelial marker *Erg* was not changed (Fig. 3a).

GATA2 protein expression was further assessed by immunostaining of transverse cryo sections of E11 embryos. Staining intensity was quantified after masking the *Prox1-GFP* signal to extract the endothelial GATA2 and VEGFR3 signals. As previously reported[20,31], VEGFR3 was highly expressed in LECs outside of the CV, showing a 2-fold increase in signal intensity compared to ECs within the CV (Fig. 3b). GATA2 immunostaining intensity was 1.9-fold higher in *Prox1-GFP*+VEGFR3high LECs outside of the CV compared to *Prox1-GFP*+VEGFR3low ECs within the CV (Fig. 3c).

Together, these in vitro and in vivo data demonstrate that GATA2 mRNA and protein levels are increased upon exposure of LECs to a soft matrix and when leaving the CV, which makes GATA2 a candidate gene to regulate early lymphatic vascular development.

**GATA2 is required for LEC emigration from the veins**. After migrating out of the CV, LECs assemble in the jugular region of the embryo into the first primitive vessels, the primordial thoracic ducts (pTD) and the peripheral longitudinal lymphatic vessels (PLLV)[20], from which most peripheral lymphatic vascular beds form by vessel sprouting. In order to investigate the potential role of GATA2 in these early steps of lymphatic development, we deleted it in the BECs prior to LEC commitment using the *Tie2-Cre* mice in combination with the floxed *Gata2* allele (Fig. 3d). Analysis of immunostained transverse vibratome sections of E12.5 embryos revealed PROX1+ pTD and PLLV (commonly referred to as 'jugular lymph sac')[20], that were consistently

smaller in *Gata2flox/flox;Tie2-Cre* in comparison to control embryos (Fig. 3e, Supplementary Fig. 2a). Light-sheet microscopy analysis of whole embryos further revealed rudimentary or malformed pTD and PLLV, and reduced number of superficial lymphatic sprouts in *Gata2* deficient embryos (Fig. 3f, Supplementary Fig. 2b, Supplementary Movies 1 and 2). All mutant

embryos analyzed (*n* = 4) also showed abnormal or absent dual lymphovenous valves and blood inside the pTD (Fig. 3f, Supplementary Fig. 2b). Consistent with reduced initial sprouting of LECs, lymphatic vessels were absent from the cervical skin of E13.5 *Gata2flox/flox;Tie2-Cre* embryos (Fig. 3g). Although early blood vessel formation appeared to occur normally in the mutant

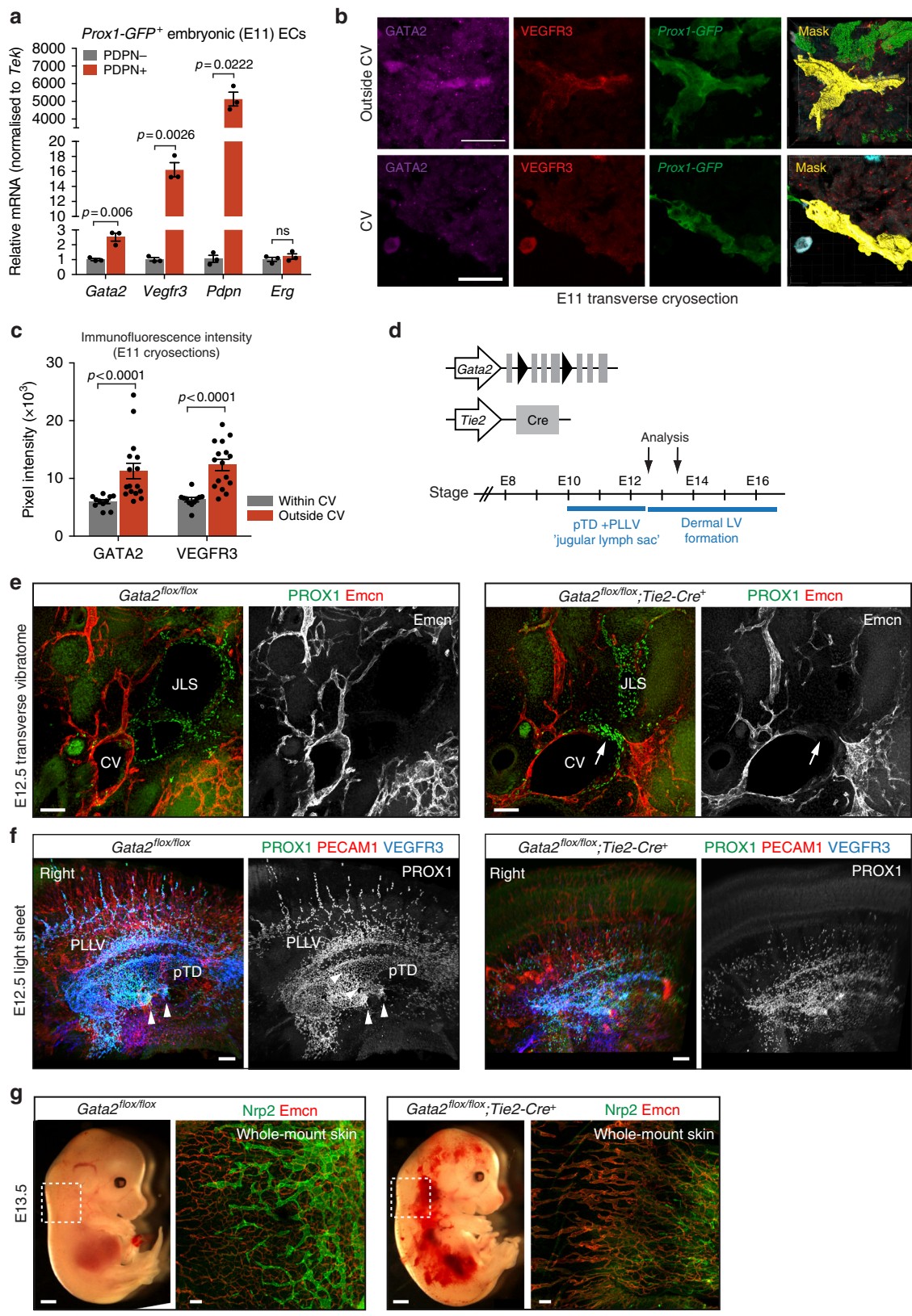

**Fig. 3** Defective migration of venous-derived LEC progenitors upon loss of endothelial *Gata2* expression. **a** Relative expression levels of *Gata2*, the LEC markers *Vegfr3* and *Pdpn* and the pan-endothelial marker *Erg* in ECs within the CV (*Prox1-GFP*[+]PDPN[−]) compared to LECs outside of the CV (*Prox1-GFP*[+]PDPN[+]). ECs were freshly isolated from E11 *Prox1-GFP* embryos (n = 3 independent samples with 4 pooled embryos in each, from two different litters). Horizontal lines represent mean ± s.e.m. *p* value, unpaired Student's *t*-test. **b** Immunofluorescence of transverse cryosections of E11 embryos using antibodies against GFP (green; *Prox1-GFP*), GATA2 (magenta) and VEGFR3 (red). Single channel images are shown. The IMARIS surface mask depicts in yellow the surface area used to extract GATA2 and VEGFR3 signal intensity. The surface mask was generated based on *Prox1-GFP* expression of VEGFR3[+] LECs within and outside of the CV. **c** Quantification of GATA2 and VEGFR3 staining intensity in ECs within (n = 35) and outside (n = 37 measurements) of the CV. Data represent mean ± s.e.m. *p* value, unpaired Student's *t*-test. **d** Schematic of the genetic constructs and analyzed embryonic stages. The time frames for pTD + PLLV (i.e. 'jugular lymph sac') and dermal lymphatic vessel formation are indicated. **e** Immunofluorescence of transverse vibratome sections of E12.5 embryos using antibodies against PROX1 (green) and Emcn (red; marker of venous EC). Single channel images for Emcn staining are shown. Note smaller jugular lymph sac (JLS) and the presence of PROX1[+] Emcn[−] LECs (arrowhead) within the CV in the mutant embryo. **f** Maximum intensity projections of E12.5 embryos (sagittal view) stained whole-mount for indicated proteins and imaged using light sheet microscopy. Single channel images for PROX1 staining are shown. Arrowheads point to lymphovenous valves that are deformed in the mutant embryos. **g** E13.5 embryos (panels on the right) and whole-mount upper thoracic dorsal skin stained for Nrp2 (green) and Emcn (red). Nrp2[+] dermal lymphatic vessels are absent in the mutant. Scale bars: 20 μm (**b**) 100 μm (**e**, **f**, **g** skins), 1 mm (**g**, embryos)

embryos (Fig. 3e, e), at E13.5 *Gata2*[flox/flox];*Tie2-Cre* showed hemorrhaging and dilation of dermal blood vessels (Fig. 3g). They died by E14.5 likely due to a combination of defects in both blood and lymphatic vessels, as well as the hematopoietic compartment that is also targeted by the *Tie2-Cre* transgene.

Accumulation of PROX1[+] cells within and adjacent to cardinal veins suggested arrested LEC migration in *Gata2* deficient embryos (Fig. 3e, Supplementary Fig. 2b). Interestingly, *Gata2-*[flox/flox];*Tie2-Cre* embryos showed normal expression of the master LEC fate regulator PROX1 in the pTD/PLLV (Fig. 3e, e, Supplementary Fig. 2b), as well as downregulation of BEC markers including Endomucin (Emcn) (Fig. 3e). This demonstrates that loss of *Gata2* does not prevent LEC differentiation but inhibits migration of venous-derived LEC progenitors.

**GATA2 regulates a stiffness-induced transcriptional program.** We next sought to analyze the role of GATA2 in regulating the matrix stiffness-induced response in LECs by comparing the transcriptomes of human LECs seeded on stiff or soft matrices in the presence or absence of GATA2 (Fig. 4a). The expression of 406 (27%) of the 1485 transcripts that were increased and 207 (16%) of the 1286 transcripts that were decreased on soft matrix was affected by GATA2 depletion (Fig. 4a). As expected[13], microarray analysis (Fig. 4b), and verification by RT-qPCR from two independent experiments (Fig. 4c), showed that exposure of LECs to a soft matrix induced downregulation of proliferation markers (*CCNB1, CCNB2, CCNA2*) and the established YAP/TAZ targets (*CTGF, ANKRD1*). Gene Ontology (GO) analysis further demonstrated soft matrix induced down-regulation of genes encoding regulators of the cell cycle and cell division (Fig. 4d), but enrichment of genes encoding molecules involved in cell–matrix adhesion and junctional organization (Fig. 4d). LECs grown on soft matrix also showed increase in GATA2-regulated expression of genes involved in cell migration (GO terms highlighted in red in Fig. 4d) and vascular development, including the known lymphatic regulators *FLT4/VEGFR3*[32], *EFNB2*[33], *ITGA9*[34], and *NOTCH1*[35,36] and its targets *HEY1* and *HEY2* (Fig. 4b). GATA2 regulated GO terms and their associated genes are depicted in Supplementary Fig. 3. Array data indicated that mRNA levels of the endothelial junctional molecules VE-cadherin and Claudin 5 were not altered. Verification by RT-qPCR from two independent experiments confirmed GATA2-dependent regulation of selected TOP genes, as indicated by abrogated or reduced soft matrix induced up- or downregulation of selected genes in GATA2 siRNA treated samples (Fig. 4e). Proliferation markers (*CCNB2, MIK67, MYBL2, BUB1*) and YAP/TAZ target genes (*CTGF, CYR61*) were not regulated by GATA2 (Supplementary Fig. 3b). In addition, although reduced baseline expression of YAP/TAZ target *ANKRD1* in GATA2 siRNA

treated samples suggests its general regulation by GATA2, soft matrix-induced downregulation still occurred (Supplementary Fig. 3b). In summary, these results indicate an important role of GATA2 in regulating soft matrix-induced transcriptional response controlling migration, but not the general mechanoresponse in LECs.

**GATA2 controls LEC responsiveness to VEGF-C.** To study the GATA2 regulated transcriptional program in LECs in vivo, we conditionally deleted *Gata2* in mouse using the tamoxifen-inducible *Vegfr3-CreER*[T2] line that efficiently targets both venous and non-venous-derived dermal LECs[37]. Gene deletion was induced prior to initiation of dermal lymphatic vessel formation by administration of 4-hydroxytamoxifen (4-OHT) from E11.5. Quantitative RT-PCR analysis of LECs isolated by flow cytometry from dorsal skins of E15.5 embryos showed downregulation of the identified matrix stiffness regulated genes *Vegfr3*, *Efnb2* and *Fgfr3* in *Gata2*[flox/flox];*Vegfr3-CreER*[T2] mice (Fig. 5a, Supplementary Fig. 4).

Next, we focused on the GATA2-dependent regulation of the major lymphangiogenic growth factor receptor VEGFR3, the activity of which has to be tightly controlled to ensure normal lymphatic development[38]. In agreement with the array data, qRT-PCR analysis showed that increased *GATA2* expression in human primary LECs seeded on soft in comparison to stiff matrix was associated with a concomitant increase in *VEGFR3* mRNA expression (Fig. 5b, c). This was dependent on *GATA2*, since *GATA2* silencing abrogated soft matrix-induced *VEGFR3* upregulation (Fig. 5c).

To determine if GATA2 directly regulates VEGFR3 expression, we performed ChIPseq in human dermal LECs. One pronounced peak within a region covering the *FLT4* gene (encoding VEGFR3) and up to 50 kb upstream of the *FLT4* promoter was identified (Fig. 5d, Supplementary Fig. 5). The GATA2 binding peak mapped to intron 1 of *FLT4* and co-localized with two indicators of active enhancer elements, H3K27Ac peak and a DNase hypersensitivity site (Fig. 5d). These data suggest that GATA2 regulates VEGFR3 expression in LECs by directly binding to a site in intron 1 of *FLT4*.

To assess the functional consequence of *Gata2* loss on the responsiveness of LECs to the VEGFR3 ligand VEGF-C, we employed an in vitro sprouting assay. We isolated primary mouse dermal LECs isolated from the *Gata2*[flox/flox];*Vegfr3-CreER*[T2] mice that showed successful in vitro deletion of *Gata2* and a decrease in the baseline expression levels of VEGFR3 upon 4-OHT treatment (Fig. 5e). VEGF-C induced sprout formation in vehicle treated control cells while *Gata2*-deleted LECs failed to sprout in response to VEGF-C (Fig. 5f–h).

Together, these results demonstrate that matrix stiffness induced increase in GATA2 levels promotes LEC responsiveness to the major lymphangiogenic growth factor VEGF-C via regulation of VEGFR3 expression.

**GATA2 is required for dermal lymphatic vessel sprouting.** The VEGF-C/VEGFR3 pathway regulates lymphangiogenic growth

during all developmental stages[39]. We therefore asked if GATA2 has a general role in regulating VEGFR3 expression and signaling, which could underlie dysfunction of dermal lymphatic vessels in Emberger syndrome[30]. We analyzed dermal lymphatic vessel sprouting in *Gata2^flox/flox;Vegfr3-CreER^T2* mice at E15.5, when non-venous derived LECs have incorporated into vessels, and at E17.5, when the laterally extending vessel sprouts reach the dorsal

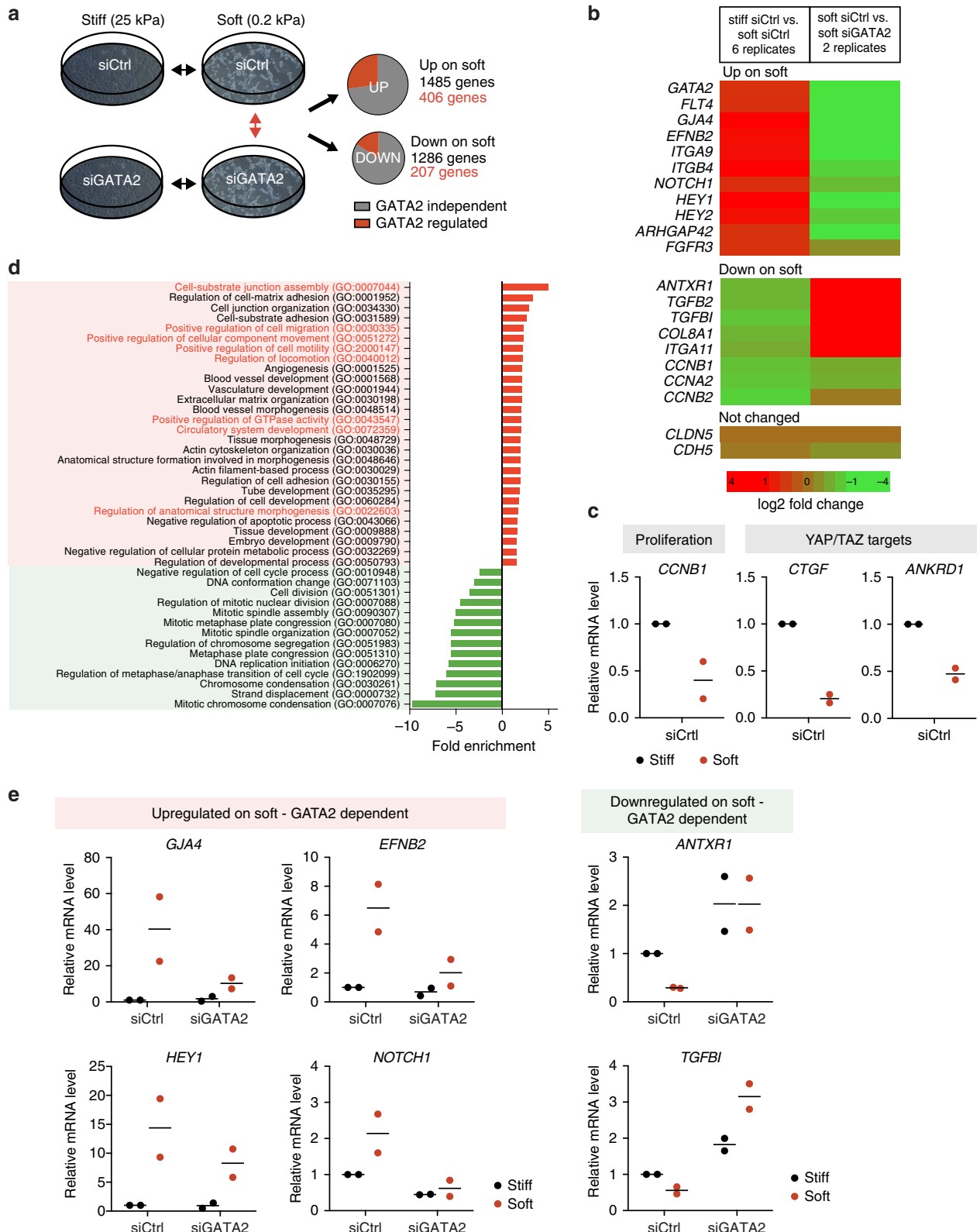

midline[37] (Fig. 6a). As expected, *Gata2*-deficient lymphatic vessels showed downregulation of VEGFR3 at both stages analyzed (Fig. 6b). At E15.5, vessel diameter in the mutants was reduced (Fig. 6d) while vessel branching, which showed a high inter-individual variability likely due to rapid development of the vasculature at this stage, was not altered (Fig. 6e). E17.5 *Gata2* deficient embryos exhibited a failure in the extension of lymphatic sprouts to the dorsal midline (Fig. 6c, f). The mutant embryos also exhibited decreased vessel branching with a concomitant increase in the diameter of lymphatic vessels (Fig. 6d,e), suggesting a defect in LEC migration. In agreement with this, *Gata2* deficient sprouts displayed a blunted morphology and reduced filopodia numbers and length (Fig. 6g-i). As previously reported[18], *Gata2* deficient vessels also lacked lymphatic valves that develop during late stages of embryogenesis (Fig. 6c). Neither the total number of LECs (Fig. 6j), nor the expression of proliferation markers *Mki67* and *Ccnb1* (Fig. 6k) was altered in the *Gata2* mutant skin and sorted dermal LECs, respectively, in comparison to control littermates. These data suggest that defective LEC migration but not proliferation underlies lymphatic vascular defects in the *Gata2*-deficient embryos.

We further investigated the LEC autonomous function of GATA2 by utilizing the *Prox1-CreER^T2* line (Supplementary Fig. 6a). Unlike the *Gata2^flox/flox*;*Vegfr3-CreER^T2* embryos, the majority (3 out of 5) of the *Gata2^flox/flox*;*Prox1-CreER^T2* embryos showed subcutaneous edema (Supplementary Fig. 6b). 2 out of 3 edematous embryos also showed blood inside the dermal lymphatic vessels. This is likely due to highly efficient *Prox1-CreER^T2* mediated targeting of the lymphovenous valves[34] that are critical for preventing blood from entering the lymphatic system[40,41] and are affected by *Gata2* loss[18] (Fig. 3c). Similar to the *Gata2^flox/flox*;*Vegfr3-CreER^T2* embryos at E17.5, *Gata2^flox/flox*; *Prox1-CreER^T2* showed decreased vessel branching, increased vessel diameter, and blunted vessel sprouts with reduced number of filopodia (Supplementary Fig. 6c-f). The earlier onset of the phenotype in the *Prox1-CreER^T2* in comparison to the *Vegfr3-CreER^T2* model may reflect different efficiencies of gene deletion. However, non-LEC autonomous effects due to edema and blood filling of lymphatic vessels in the *Gata2^flox/flox*;*Prox1-CreER^T2* embryos cannot be ruled out.

Taken together, these results demonstrate a LEC-autonomous function of GATA2 in modulating lymphangiogenic growth factor responsiveness of LECs and controlling dermal lymphatic vessel sprouting and patterning.

**LECs respond differently to matrix stiffness and OSS**. The above results reveal matrix-stiffness induced activation of a GATA2-regulated transcriptional program in LECs in response to decreased mechanical stimulus (soft matrix). Conversely, increased mechanical stimulus triggered by oscillatory flow, which provides a signal for the initiation of valve morphogenesis at vessel branch points[16], increases GATA2 levels and GATA2-dependent expression of key regulators of valve morphogenesis[18].

In order to define whether the transcriptional control mechanisms of these GATA2-dependent mechanosensitive steps of lymphatic vascular morphogenesis are shared, we compared stiffness-induced gene expression changes on soft matrix to those previously shown to be induced by oscillatory shear stress (OSS)[14]. Consistent with the observed cell cycle arrest by both stimuli (Fig. 4c)[14,42], we found among downregulated transcripts enrichment of genes involved in cell proliferation, including those encoding the Cyclin and Kinesin superfamily proteins (Fig. 7a). However, only a small proportion of genes overall were regulated similarly by soft matrix and OSS. GO analysis of the upregulated transcripts showed enrichment of genes controlling vascular development and cell migration. Notably, matrix metalloproteinase (MMP) signalling components (MMP1, MMP2 and MMP10), implicated as positive regulators of lymphangiogenesis[43,44], were upregulated both by exposure of LECs to soft matrix or OSS. OSS, but not soft matrix exposure also led to highly significant enrichment of upregulated genes associated with metabolic processes.

Differential transcriptional responses to soft matrix and OSS exposure, although partly controlled by GATA2, may not be entirely unexpected given the opposite effects of the two stimuli on the general YAP/TAZ dependent mechanotransduction pathway. We therefore asked if the transcriptional responses to increased mechanical stimuli by stiffer matrix and OSS are shared. As expected, LECs exposed to stiff matrix or OSS showed increased expression of the YAP/TAZ target gene *CTGF* (Supplementary Fig. 7). However, only a small subset of high stiffness-regulated genes (8%), mostly implicated in cell adhesion and locomotion, showed a similar down- or upregulation, respectively, by OSS, suggesting unique mechanoresponses in LECs to different stimuli.

## Discussion

Tissue mechanics are known to regulate morphogenetic processes, but the effect of ECM stiffness on vascular development, and how mechanical forces resulting from differential matrix stiffness translate into intracellular signals in ECs are poorly understood. In this study, we uncover a role for the GATA2 transcription factor in controlling lymphangiogenic growth factor responsiveness in venous-derived LECs as they move from a stiffer environment in the cardinal vein wall to a softer matrix outside of the cardinal vein. We further identify an important role of GATA2 in regulating baseline VEGFR3 expression levels, and thereby LEC migration and lymphatic vessels growth, also at later stages of development.

Early stages of lymphatic development require PROX1 and VEGF-C/VEGFR3 that regulate LEC differentiation and migration out of the veins, respectively[31,45]. A positive feedback loop between PROX1 and VEGFR3 was identified as a key mechanism controlling the identity and number of venous-derived LEC progenitors[46]. Our data show that in the absence of GATA2, PROX1 is not sufficient to confer LEC responsiveness to VEGF-C

**Fig. 4** GATA2 regulates a matrix stiffness-induced transcriptional program in the LECs. **a** Schematic overview of microarray gene expression analysis comparing Ctrl and *GATA2* siRNA-treated human LECs grown on stiff (25 kPa) or soft (0.2 kPa) matrix. Number of genes regulated by matrix stiffness ($n = $ 6 biological replicates) and the proportion of those showing GATA2-regulated expression ($n = 2$ biological replicates) are shown. **b** Selected TOP genes that are upregulated or downregulated on soft (0.2 kPa) in comparison to stiff (25 kPa) matrix and affected by GATA2 depletion. Heatmap color coding shows maximum for log2 fold change >1 or <−1. **c** Verification of microarray data in 2 additional independent experiments by qRT-PCR analysis shows the expected downregulation of markers of proliferation (*CCNB1*) and YAP/TAZ dependent mechanosignalling (*CTGF, ANKRD1*) in LECs grown on soft matrix. **d** Selected significantly enriched GO terms in LECs grown on soft matrix. The fold enrichment of upregulated (red bars) and downregulated (green bars) is shown. Genes associated with cell–matrix adhesion, cell migration and vascular development are upregulated on soft matrix, whereas genes associated with cell proliferation are downregulated. Notably, GO terms associated with cell migration, locomotion and motility (highlighted in red) are upregulated in a GATA2 dependent manner. **e** Verification of selected matrix stiffness regulated genes from the microarray expression analysis by qRT-PCR analysis in two independent experiments. Horizontal lines represent mean

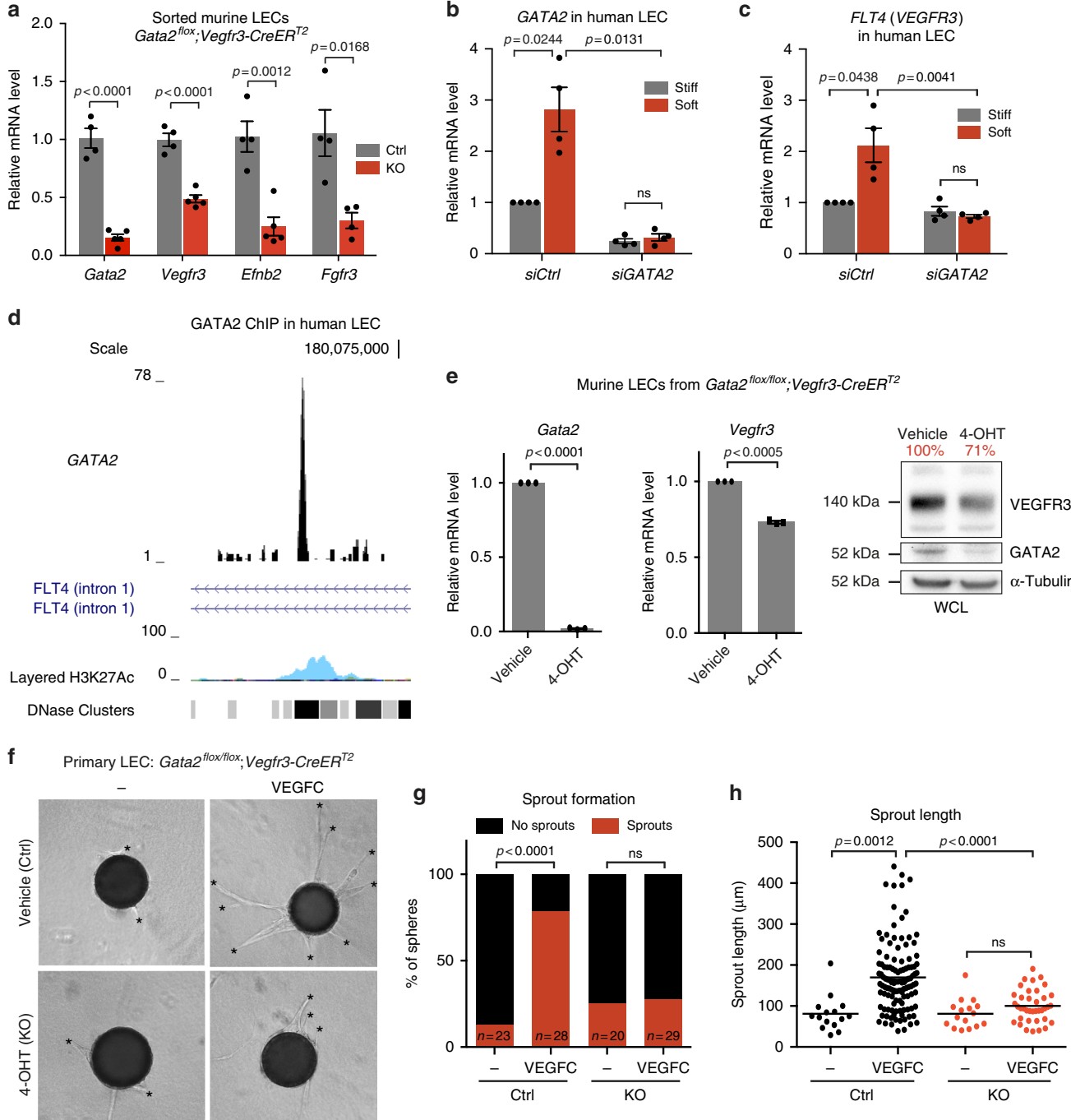

**Fig. 5** GATA2 regulates VEGFR3 expression and LEC responsiveness to VEGF-C. **a** Relative mRNA expression levels of *Gata2*, *Vegfr3, Efnb2* and *Fgfr3* in freshly isolated LECs from E15.5 *Gata2* mutant ($n = 4$ or $n = 5$) and Cre[−] littermate control ($n = 4$) embryos. Horizontal lines represent mean ± s.e.m. *p* value, unpaired Student's *t*-test. **b**, **c** qRT-PCR analysis of *GATA2* and *VEGFR3* in control (Ctrl) and GATA2 siRNA-treated human LECs grown on soft (0.2 kPa) or stiff (25 kPa) matrix. $n = 4$ experiments, mean ± s.e.m. *p* value, one-sample *t*-test (for siCtrl stiff vs soft) or unpaired Student's *t*-test. **d** Occupancy of chromatin at the *FLT4* (encoding VEGFR3) locus as viewed in UCSC Human Genome Browser (http://genome.ucsc.edu/). GATA2 ChIP-seq profile demonstrating binding at the first intron of the *FLT4* locus in human dermal LECs. Co-occupancy with H3K27Ac and sites of DNase hypersensitivity, both marks of active enhancer elements, are shown in ENCODE tracks (HUVEC, light blue peaks). **e** qRT-PCR and western blot analysis of *Gata2* and *Vegfr3* in control and 4-OHT treated (*Gata2* deleted) primary mouse LECs. For qRT-PCR: $n = 3$ experiments, mean ± s.e.m. *p* value, one-sample *t*-test. Main band (125 kDa) quantification is shown in red. **f** Vehicle (Ctrl) and 4-OHT (KO) treated primary mouse LEC spheres in the absence (−) or presence of VEGF-C. Asterisks indicate sprouts. **g** Proportion of control (Ctrl) and *Gata2* deficient (KO) LEC spheres forming sprouts in the absence (−) or presence of VEGF-C. n is indicated; *p* value, Fisher's exact test. **h** Quantification of sprout length. Dots represent individual sprouts. Horizontal lines represent mean. $n = 20$–30 beads (from two independent experiments). *p* value, unpaired Student's *t*-test

in vitro or in vivo. On the other hand, GATA2 function was not required for PROX1 expression and LEC differentiation. The GATA2-dependent regulation of genes involved in cell migration and lymphangiogenesis, including VEGFR3, but not those encoding general regulators of LEC mechanoresponse to soft matrix, points to a specific role of GATA2 in controlling the migratory behavior of

LECs. In support of this, we show that GATA2 is required in vivo for the development of two lymphatic vessel networks that form through a process of LEC migration and/or vessel sprouting; the first primitive lymphatic vessels in the jugular region[20,21] of the embryo and the lymphatic vasculature of the dorsal skin[37]. In the mesentery, where lymphatic vessels form by lymphvasculogenic

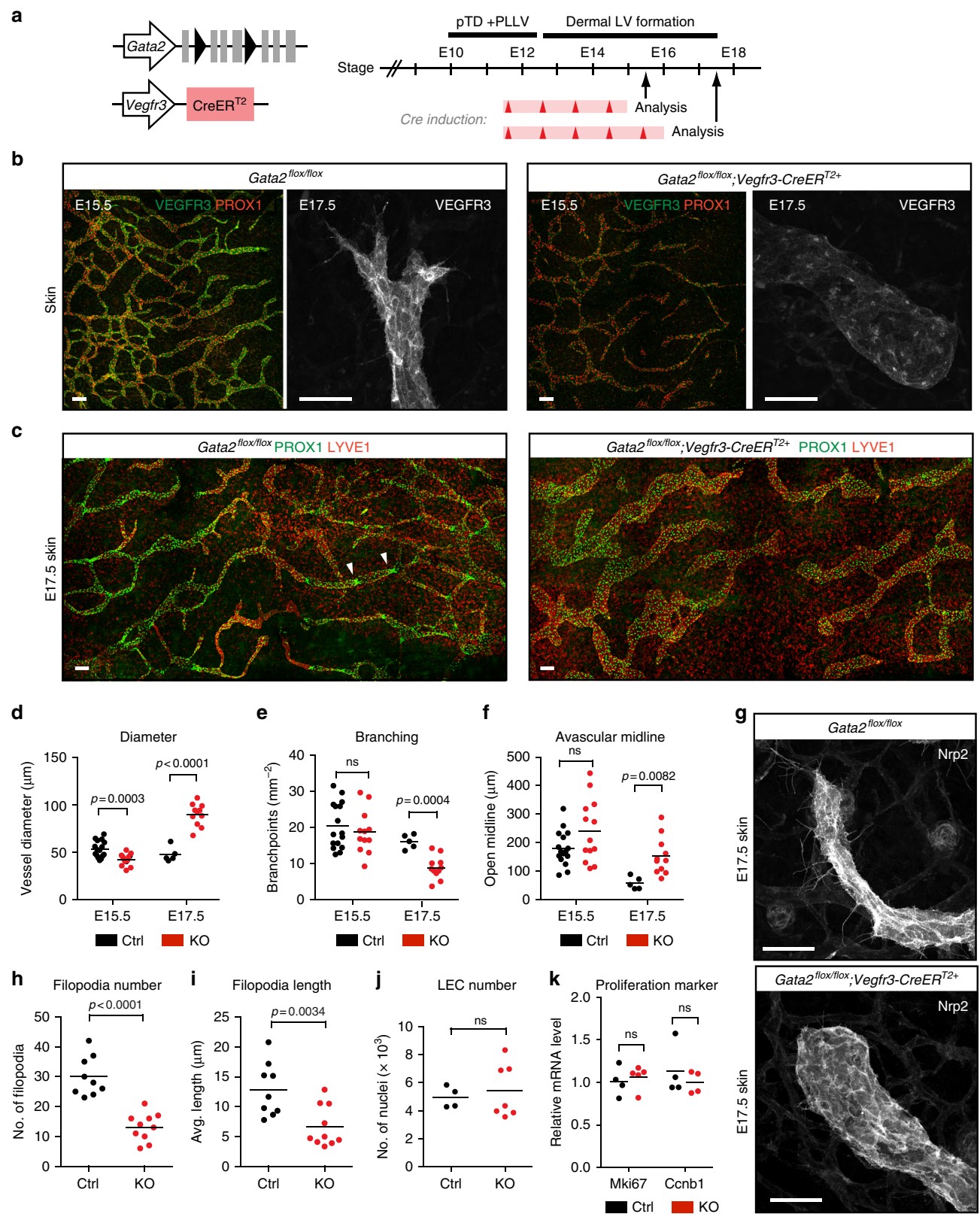

**Fig. 6** LEC autonomous function of GATA2 in dermal lymphatic vessel sprouting and patterning. **a** Schematic of the genetic constructs, 4-OHT treatments (red arrowheads; red bars represent expected time frame of 4-OHT activity) and analyzed embryonic stages. The time frames for pTD + PLLV and dermal lymphatic vessel formation are indicated. **b** Whole-mount immunofluorescence of E15.5 and E17.5 upper thoracic dorsal skins using antibodies against VEGFR3 (green) and PROX1 (red), or VEGFR3 alone (single channel images on the right). Note reduced VEGFR3 levels in the mutant. **c** Whole-mount immunofluorescence of E17.5 upper thoracic dorsal skins using antibodies against PROX1 (green) and LYVE1 (red). Note developing valves consisting of clusters of highly PROX1 positive cells (arrowheads) that are present in control skin only. **d**–**f** Quantification of lymphatic vessel diameter (**d**), branch points (**e**) and the distance between contralateral sprouts (open midline; **f**) in E15.5 and E17.5 dorsal skin in *Gata2* mutant (KO) and Cre⁻ littermate control embryos. Horizontal lines represent mean ($n = 16$ (E15.5 Ctrl), $n = 12$ (E15.5 KO), $n = 5$ (E17.5 Ctrl), $n = 11$ (E17.5 KO)). *p* value, unpaired Student's *t*-test. **g** Morphology of lymphatic vessel sprouts in E17.5 skin. **h**, **i** Quantification of filopodia numbers (**h**) and length (**i**) in E17.5 dermal lymphatic vessel sprouts. Horizontal lines represent mean ($n = 9$ (Ctrl) and $n = 10$ (KO) embryos), *p* value, unpaired Student's *t*-test. **j** Quantification of LEC nuclei in E17.5 dorsal skin. Horizontal lines represent mean ($n = 4$ (Ctrl) and $n = 7$ (KO) embryos). *p* value, unpaired Student's *t*-test. **k** Relative mRNA expression levels of the proliferation markers *Mik67* and *Ccnb1* in dermal LECs sorted from E15.5 *Gata2* mutant (KO; $n = 4$ or $n = 5$) and Cre⁻ littermate control (Ctrl; $n = 4$) embryos. Horizontal lines represent mean ± s.e.m. *p* value, unpaired Student's *t*-test. Scale bars: 100 μm (double-stained images in **b**, **c**), 50 μm (single channel images in **b**, **g**)

assembly from progenitors[47], initial vessel formation is not affected but valve morphogenesis is inhibited[18].

Intriguingly, as opposed to the previously reported activation of GATA2 in ECs in response to increased mechanical stimulus[17,18], we found that GATA2 expression is increased in LECs upon exposure to a soft matrix and when LECs have left the CV (i.e. decreased mechanical stimulus and YAP/TAZ signalling). Moreover, our data suggest that the mechanosensitive transcriptional program activated in LECs in response to different mechanical stimuli, namely increased matrix stiffness and oscillatory flow are remarkably different. It has been described in several cell types including ECs that GATA2 interacts with other transcriptional regulators, including Etv2[48] and Lmo2[49], to form multimeric transcription complexes. An interesting possibility that should be addressed in future studies is whether oscillatory flow- and soft matrix-induced differences in the GATA2 mediated regulation of target genes and cellular responses is explained by formation of different transcriptional complexes. Interestingly, while OSS induces cell cycle arrest in the LECs[14], stiff matrix promotes cell proliferation and adhesion. Yet other mechanical forces exerted by stretching[50] and laminar flow[51,52] also increase LEC proliferation. It should be noted that many of the baseline behaviors of LECs and other cell types are recorded in in vitro cultures on plastic or glass, the stiffness of which is in the gigapascal range and therefore orders of magnitude stiffer than any living tissue. For example, the stiffness of embryonic tissue, brain and liver is only 0.2–0.6 kPa ([27] and this study).

Exposure of cultured LECs to a soft matrix of 0.2 kPa, corresponding to the stiffness of embryonic tissue that provides a substrate for LEC progenitor migration in vivo, led to GATA2-dependent changes in the expression of several known lymphatic regulators. Besides VEGFR3, these genes may also contribute to the lymphatic vascular defects in the *Gata2* deficient mice. For example, *FGFR3* that controls early lymphatic vessel development[53] was upregulated in LECs grown on soft matrix, and downregulated in *Gata2* deficient LECs in vivo. TGFβ2 was similarly downregulated on soft matrix in a GATA2-dependent manner, consistent with the previously reported regulation of TGFβ1 by GATA2 during erythroid differentiation[54]. TGFβ signalling has a context-dependent role in lymphatic vasculature and has been shown to both promote[55] and inhibit[54] lymphangiogenesis. A similar bi-functional role during lymphangiogenic growth has been described for NOTCH signalling. At early stages of development, NOTCH1 activity limits excessive LEC proliferation[35,56]. In agreement with this, we found that LECs grown on soft matrix showed GATA2-dependent upregulation of NOTCH1 and its downstream targets HEY1/2. GATA2-dependent, soft matrix-induced upregulation of genes controlling valve morphogenesis, including EphrinB2[33], integrin α9[57] and Connexin 37[58] may further indicate previously unrecognized

functions of these genes also during the early steps of lymphatic development.

Cyclic stretching of cells cultured on soft matrix recapitulates the phenotype of cells grown on stiff matrix, namely stress fiber formation, cell spreading and proliferation[59]. Interestingly, LEC stretching, caused by increase in interstitial fluid pressure between E11.5 and E12.0 of mouse development, was previously shown to promote integrin ß1-dependent VEGFR3 phosphorylation and LEC proliferation[50]. This, together with the data presented in our study, suggests a stepwise regulation of early lymphatic vascular expansion by biomechanical signals. First, upon exiting the cardinal vein, exposure of LECs to a soft matrix leads to cell cycle exit and upregulation of genes involved in cell migration (Fig. 7b). Importantly, GATA2 mediated VEGFR3 upregulation is critically required for directed VEGF-C-dependent migration of LECs to establish the primary network of LECs between E10.5-E11. In the second step (E11.5), increase in the interstitial fluid pressure and concomitant stretching of LECs stimulate VEGFR3 phosphorylation, LEC proliferation and vessel growth[50]. In the third step (E11.5 onwards), flow induced mechanical forces further shape the developing vasculature. Initiation of (laminar) flow in the vessels has been shown to promote vessel sprouting through inhibition of NOTCH1, while KLF2/4 induces proliferation[51,52]. Later (from E15 onwards), turbulent flow at vessel branch points within a primitive vascular plexus induces a FOXC2 and GATA2-dependent LEC quiescence and valve endothelial phenotype[14,18,19].

Tissue stiffness increases during embryonic development and is linked to gradual organization and enrichment of collagen fibers[4,60,61]. In addition to general stiffening of the tissue, local changes in matrix stiffness are likely of critical importance. It is interesting to note that soft, but not stiff matrix has been shown to precisely regulate cellular functions such as adhesion and invasiveness in a scale range of 0.1 kPa[62], suggesting that small changes in stiffness in soft (e.g. embryonic) tissues can have profound effects on cellular behaviors. In the vasculature, the deposition and assembly of the basement membrane on basal endothelial surfaces is likely to dramatically change the mechanical properties of their substrate. It is tempting to speculate that migrating tip cells leading the vascular sprouts experience a softer matrix when not supported by the basement membrane of the vessel wall. Notably, VEGFR3 is highly expressed in the blood endothelial tip cells[63]. However, we do not have evidence for stiffness-regulated expression of GATA2 and VEGFR3 in either blood or lymphatic vessel sprouts. The ability of GATA2 to regulate baseline levels of VEGFR3 expression may instead explain dermal lymphatic vessel sprouting defects in the *Gata2* deficient embryos.

Previous work identified inactivating mutations in the *GATA2* gene as causative of Emberger syndrome, which is an autosomal

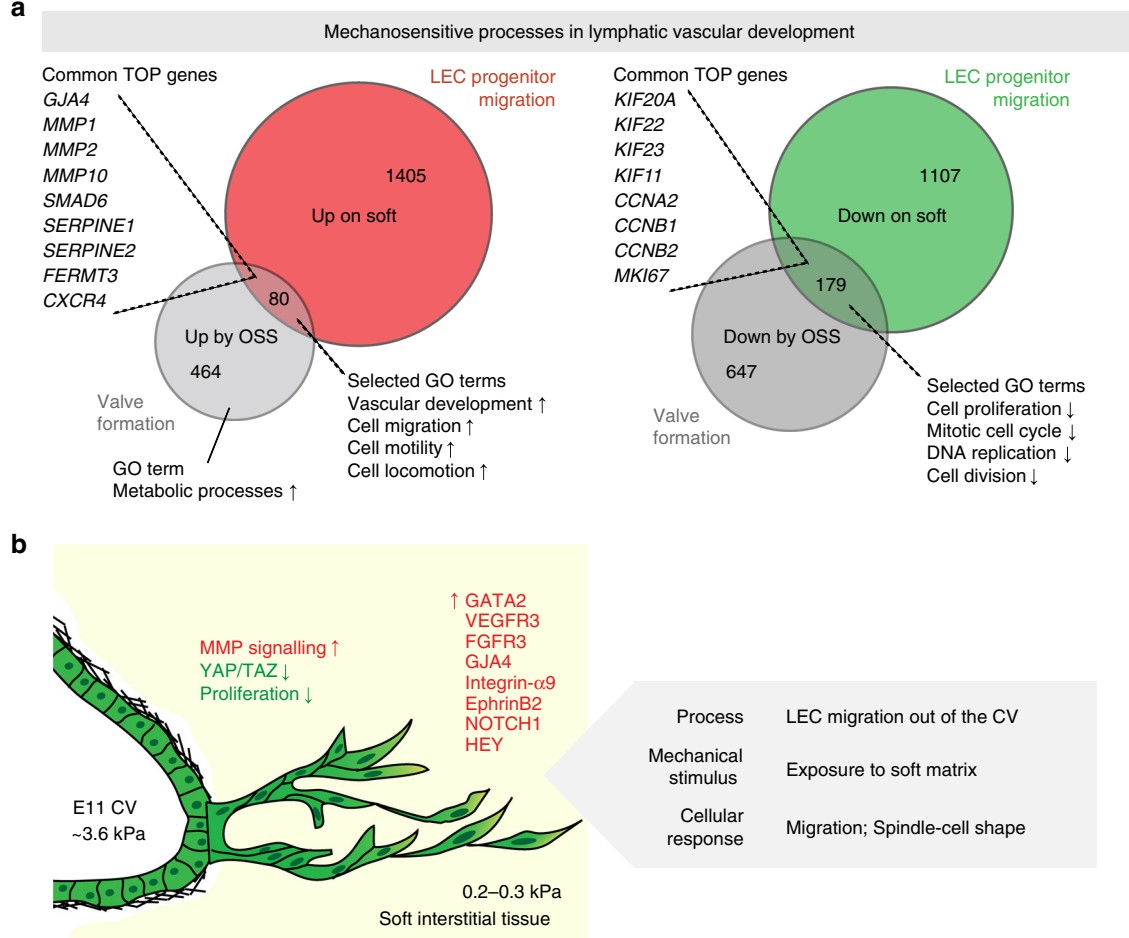

**Fig. 7** Common and unique mechanoresponses during lymphatic vascular morphogenesis. **a** Area-proportional Venn diagrams of up- or downregulated genes in LECs grown on stiff or soft matrix (mimicking conditions of venous-derived LEC migration during the formation of first lymphatic vessels), and LECs subjected to static or oscillatory flow (mimicking initiation of valve formation[14,16]) conditions. Common TOP genes and selected GO terms are listed. **b** Schematic of venous-derived LEC migration out the CV. Exposure of venous-derived LECs to soft matrix induces GATA2-dependent increase in VEGFR3 expression and LEC migration. TOP regulated genes (red, up; green, down) are indicated

dominant condition with lymphedema in lower limbs and genitalia[29,30]. Studies in mice further demonstrated a specific role of GATA2 in the formation of lymphatic and lymphovenous valves[18]. Lymphatic valve defects are, however, unlikely to fully explain the early-onset lymphedema and the lack of lymphatic uptake of intradermally injected tracer in patients with Emberger syndrome[30]. Given that a large proportion (>30%) of proteins encoded by hereditary lymphedema-associated genes act on the VEGF-C-VEGFR3 signalling pathway[64], our finding that GATA2 regulates VEGF-C responsiveness of LECs may provide an explanation for lymphatic vascular failure caused by GATA2 loss in Emberger syndrome. It is also interesting to note that chronic (lymph)edema is often accompanied by tissue stiffening[65]. Our results further suggest that targeting tissue stiffness is important for controlling disease progression and response to (lymph) angiogenic therapy.

In summary, we demonstrate a hitherto unrecognized role of ECM stiffness in regulating vascular morphogenesis in vivo. We further uncover that the mechanosensitive transcriptional program activated in LECs in response to matrix stiffness is different from that induced by oscillatory shear. Given that all cells are exposed to many different types of mechanical forces (including shear, matrix stiffness, stretching, pulling, compression), which are increasingly recognized as key regulators of cell behavior and fate during development, homeostasis and disease, our findings

call for further investigation into identifying common and unique mechanoresponses in ECs and other cell types.

## Methods

**Mice**. *Gata2*$^{flox66}$, *Tie2-Cre*[67], *Prox1-CreER*$^{T268}$, *Vegfr3-CreER*$^{T269}$ and *Prox1-GFP*[37] mice were previously described. *Tie2-Cre*$^+$ male mice (*Gata2*$^{flox/+}$;*Cre*$^+$) were crossed with *Gata2*$^{flox/flox}$;*Cre*$^-$ females to avoid inheritance of a null allele when transmitted through the female germ line. For induction of Cre, 4-OHT was dissolved in peanut oil (10 mg ml$^{-1}$) and administered to pregnant females by intraperitoneal injection. For *Gata2* gene deletion using *Vegfr3-CreER*$^{T2}$ and *Prox1-CreER*$^{T2}$ lines and analysis at embryonic day (E)15.5, four consecutive injections of 1 mg of 4-OHT were administered daily starting at E11.5. For analysis at E17.5, an additional injection at E15.5 was performed. The same phenotype was observed in the *Gata2*$^{flox/flox}$;*Vegfr3-CreER*$^{T2}$ embryos using three consecutive injections of 2 mg of 4-OHT at E11.5, E12.5 and E13.5. The morning of vaginal plug detection was considered as embryonic day 0. All strains were maintained and analyzed on a C57BL/6 J background (backcrossed minimum 6 times). Experimental procedures were approved by the United Kingdom Home Office or the Uppsala Laboratory Animal Ethical Committee.

**Cell culture and in vitro gene deletion using 4-OHT**. Human primary dermal lymphatic endothelial cells (HDLEC from juvenile foreskin, cat. C12216) were obtained from PromoCell. Cells were seeded on bovine Fibronectin-coated dishes in complete ECGMV2 medium (PromoCell) supplemented with 25 ng ml$^{-1}$ of VEGF-C (R&D Systems). Murine primary dermal lymphatic endothelial cells were isolated from 10 weeks old *Gata2*$^{flox/flox}$;*Vegfr3-CreER*$^{T2}$ females. Tails were cut at the base and transferred into sterile HBSS + 1% penicillin and streptomycin (PS) before the skin was removed from the tails and cut into 30 mm long fragments. The fragments were incubated in 5% dispase (Roche) in PBS for 45 min at 37 °C with

gentle agitation to separate epidermis from dermis. Dermal fragments were washed in HBSS + 1% PS, transferred into collagenase A (1 mg ml$^{-1}$) (Roche) in PBS solution, filtered and then seeded on 0.5% gelatin-coated dishes[69]. When cells reached confluence, they were detached with Accutase, incubated with rat anti-mouse PECAM1 antibody (Mec13.3, Pharmingen, 4 μg per 30 mm dish) for 45 min at 4 °C and sorted with Dynabeads conjugated with sheep anti-rat antibodies. One passage later, cells were sorted again as above using LYVE1 antibody (Aly7, Abnova, 3 μg per 30 mm dish). Pure dermal LECs were cultured on 0.5% gelatin-coated dishes and passaged 1:3 after 4 days. In vitro gene deletion was induced by adding 1–2 μM 4-OHT (Sigma) to the culture medium. All cells were cultured at 37 °C in a humidified atmosphere with 5% CO$_2$.

**RNA interference**. Stealth RNAi Duplexes and the corresponding Medium GC Stealth RNAi Control Duplexes (Invitrogen) were used to knock-down human GATA2. The GATA2 target sequences used were the following: 5′-CAGAC-GAGGTGGACGTCTTCTTCAA-3′ (siRNA #1) and 5′-CAGCAAGGCTCGTT CCTGTTCAGAA-3′ (siRNA #2). Cells were subjected to two consecutive cycles of transfection 24 h and 48 h after plating, using Lipofectamine 2000 (Invitrogen) according to the manufacturer's instructions.

**In vitro matrix stiffness analysis**. To analyze the effects of matrix stiffness, confluent LECs grown on Fibronectin-coated plastic cell culture dishes were detached with trypsin-EDTA 0.05% and plated on soft (0.2 kPa) or stiff (4, 8, 12 or 25 kPa) bovine Fibronectin-coated Softwell™ or Softslip™ dishes (Matrigen) for 24 h. For analysis of the effect of GATA2 knockdown, LECs were subjected to two consecutive cycles of transfection (day 0 and day 1), seeded on day 2 on soft or stiff matrices and analyzed 24 h later.

**LEC sprouting assay**. Confluent murine dermal LECs isolated from Gata2$^{flox/flox}$; Vegfr3-CreER$^{T2}$ mice were detached with trypsin and $1.5 \times 10^6$ cells were incubated with 3000 Cytodex-3 beads (rehydrated in PBS) for 4 h in culture medium suspension at 37 °C. Cell-coated beads were subsequently incubated for 48 h in cell culture dishes to achieve efficient coating of the bead. During this incubation deletion of Gata2 was induced by adding 1 μM of 4-OHT (or vehicle, EtOH) to the culture medium. After 48 h, 4-OHT-supplemented medium was removed and beads were embedded in a fibrin gel (4 mg ml$^{-1}$). Vehicle or 4-OHT treated LECs were cultured in the absence or presence of 200 ng ml$^{-1}$ of VEGF-C (R&D Systems) for three days, after which the beads were photographed for quantification of sprout numbers and length.

**Western blot analysis**. Total protein extract was obtained by solubilizing cells in boiling Laemmli buffer [2.5% SDS and 0.125 M Tris-HCl (pH 6.8)]. Cell lysates were incubated 5 min at 100 °C to induce protein denaturation and then spinned for 5 min at 16,100 g to discard cell debris. Supernatants were collected and protein concentration was determined using a BCA™ Protein Assay Kit (Pierce) according to manufacturer's instructions. Equal amounts of proteins were loaded on gel and separated by SDS-PAGE, transferred to a PVDF Membrane 0.2 μm pore size (BIORAD) and blocked for 1 h at RT in 1× Tris Buffered Saline Tween (TBST) [150 mM NaCl, 10 mM Tris-HCl (pH 7.4), and 0.05% Tween] containing 5% w/v powdered milk. The membranes were subjected to overnight incubation at 4 °C or 1 h at RT with primary antibodies diluted in 1× TBST-5% BSA. Following this incubation, membranes were rinsed three times with 1× TBST for 5 min each and incubated for 1 h at RT with HRP-conjugated secondary antibodies (diluted in 1× TBST-5% BSA). Membranes were rinsed three times with TBST for 5 min each and specific binding was detected using the enhanced chemiluminescence (ECL) system (Amersham Biosciences) and the Gel Doc™ XR + System. Protein molecular masses were estimated relatively to the electrophoretic mobility of co-transferred pre-stained protein marker, Broad Range (Cell Signaling Technology). The following primary antibodies were used: rabbit anti-VEGFR3 (cat. SC-321, Santa Cruz Biotechnology, 0.5 μg ml$^{-1}$), goat anti-GATA2 (cat. 4595, Cell Signaling Technology, 0.5 μg ml$^{-1}$) and mouse anti-α-Tubulin (cat. T5168, Sigma-Aldrich, 0.1 μg ml$^{-1}$). Full blots are shown in Supplementary Fig. 8.

**Flow cytometry**. Embryonic tissue from E11 Prox1-GFP$^+$ embryos or E15.5 Gata2$^{flox/flox}$;Vegfr3-CreER$^{T2}$ embryos were harvested, dissected in cold PBS and digested in 4 mg ml$^{-1}$ Collagenase IV (Life technologies), 0.2 mg ml$^{-1}$ DNaseI (Roche) and 10% FBS (Life technologies) in PBS at 37 °C for 10–15 min with vigorous shaking every 5 min. Collagenase activity was quenched by dilution with FACS buffer (PBS, 0.5% FBS, 2 mM EDTA) and digestion products were filtered twice through a 70 μm nylon filter (BD Biosciences). Cells were washed with FACS buffer and immediately processed for enrichment of CD31 (PECAM1) positive cells using magnetic beads according to the manufactures instructions (Miltenyi Biotec). After enrichment, Fc receptor binding was blocked with rat anti-mouse CD16 (CD32), (cat. 14-0161, eBioscience, 1:100). Samples were thereafter stained with anti-podoplanin PE (eF660, cat. 50-5381, 1:100) and anti-CD31 (PECAM1) (390) PE-Cy7 (cat. 25-0311, eBioscience, 1:200). Immune cells and erythrocytes as well as dead cells were excluded using anti-CD45 (30-F11, cat. 48-0451, 1:50), anti-CD11b (M1/70, cat. 48-0112, 1:50) and anti-TER-119 e450 (TER-119, cat. 48-5921, 1:50) (all from eBioscience), together with the cell death dye Sytox blue (Life

technologies), all detected by the violet laser as one dump channel. For compensation, the AbC anti-rat/hamster compensation bead kit (Life Technologies) was used. Cells were sorted on a BD FACSAria III cell sorter equipped with a 85 μm nozzle and data were processed using FlowJo software (TreeStar). Single cells were gated from a FSC-A/FSC-H plot followed by exclusion of dead cells and immune cells and erythrocytes in the violet dump channel. Subsequent gating to obtain cell fractions to be sorted is presented in Supplementary Fig. 4.

**qRT-PCR analysis**. For qRT-PCR analysis of human LECs, total RNA was isolated by RNeasy Mini kit or RNeasy Micro kit (QIAGEN) and 0.5 μg was reverse transcribed using oligo dT (SuperScript III First-Strand Synthesis System, Invitrogen) or SuperScript® VILO cDNA Synthesis Kit according to the manufacturer's instructions. For analysis of sorted mouse LECs, total RNA was extracted by RNeasy Micro kit (QIAGEN) and all obtained RNA was reverse transcribed using oligo dT (SuperScript III First-Strand Synthesis System, Invitrogen). cDNA was pre-amplified using the TaqMan PreAmp Master Mix Kit. Gene expression levels were analyzed using TaqMan Gene Expression Assay (Applied iBiosystems) and a 7900 HT Fast Real-Time PCR System thermocycler (Applied Biosystems) following manufacturer's instructions. Relative gene expression levels were normalized to GAPDH or the endothelial marker TEK. The following probes were used: Hs99999905 GAPDH, Hs00176607 FLT4, Hs00216777 ANTXR1, Hs01114113 HEY1, Hs1062014 NOTCH1, Hs01030099 CCNB1, Hs00170014 CTGF, Hs00932747 TGFBI, Hs00704917 GJA4, Hs00187950 EFNB2, Hs00173317 ANKRD1, Hs00231119 GATA2, Hs00945150 TEK, Hs01554629 ERG, Hs04260396 MIK67, Hs01084593 CCNB2, Hs00942540 MYBL2, Hs01557695_m1 BUB1, Hs00155479 CYR61, Ms99999915 Gapdh, Ms00492300 Gata2, Ms01292604 Flt4, Mm00443243 Tek, Mm01214244 Erg, Mm00438670 Efnb2, Mm00433294 Fgfr3, Mm01278617 Mki67, Mm01171453 Ccnb2.

**Microarray expression analysis**. Human LECs were treated with control siRNA or GATA2 siRNA #1 as described above, and cultured on soft (0.2 kPa) or stiff (25 kPa) matrices for 24 h. Total RNA was isolated using the RNeasy Mini kit (QIAGEN) and DNA digestion was performed using the RNase-Free DNase Set (QIAGEN). RNA concentration and quality was evaluated using the Agilent 2100 Bioanalyzer system (Agilent Technologies Inc, Palo Alto, CA). 10 ng of total RNA starting amount was used to generate amplified and biotinylated sense-strand cDNA from the entire expressed genome according to the GeneChip® WT Pico Reagent Kit User Manual (P/N 703262 Rev 1 Affymetrix Inc., Santa Clara, CA). GeneChip® ST Arrays (GeneChip® Human Transcriptome Array 2.0) were hybridized for 16 h in a rotating 45 °C incubator, according to the GeneChip® Expression Wash, Stain and Scan Manual (PN 702731 Rev 3, Affymetrix Inc., Santa Clara, CA). The arrays were then washed and stained using the Fluidics Station 450 and finally scanned using the GeneChip® Scanner 3000 7 G.

The raw data was normalized to gene level in Expression Console, provided by Affymetrix (http://www.affymetrix.com), using the robust multi-array average (RMA) method[70,71]. Subsequent analysis of the gene expression data was carried out in the statistical computing language R (http://www.r-project.org) using packages available from the Bioconductor project (www.bioconductor.org). To identify differentially expressed genes between the ctrl stiff and ctrl soft groups, a stepwise analysis with 6 biological replicates was performed. First, exon set ID's with an average expression lower than 5 were considered as not significantly expressed and excluded from the analysis. A threshold of 40% increase (>0.5 log2 fold change) or decrease (<−0.5 log2 fold change) of gene expression on the soft matrix (vs. stiff matrix) was considered for further analysis. For all genes with three or more exon probe set ID's regulated above the defined thresholds, the average log2 fold change of the regulated exon probe ID's was calculated and used to generate the final list of genes regulated by matrix stiffness. 3.8% of the regulated genes were found to be both increased and decreased, potentially indicating differential expression of different splice variants. These genes were excluded from further validations. To identify regulated transcription factors the most updated database of transcription factors was exported[72]. Additionally, we employed the TFactS online module (https://omictools.com/tfacts-tool) to predict the core transcription factors of stiffness regulation in LECs[28]. To determine which of the genes are regulated in a GATA2 dependent manner, a stepwise analysis with 2 biological replicates ctrl siRNA soft vs GATA2 siRNA soft groups was performed. The previously identified exon probe ID's for genes differentially regulated between the ctrl stiff and ctrl soft groups were extracted and their average log2 fold change was calculated for the new data set. A threshold of 40% increase (>0.5 log2 fold change) or decrease (<−0.5 log2 fold change) of gene expression in the absence of the GATA2 was used to generate the final list of genes.

To compare gene expression changes in LECs exposed to oscillatory shear stress (OSS), we utilized microarray data from a previously published study[14]. Processed array data were downloaded from NCBI GEO database (accession number: GSE60152) and the four control siRNA treated samples, representing two replicates for both static and OSS conditions, were used for analysis. Same array filtering criteria were applied as above, including the exclusion of probe sets with average expression lower than five from the analysis. The average fold changes between LEC expression levels under OSS vs static conditions were calculated. 544 genes showing increased expression (>0.5 log2 fold change) and 826 genes showing

decreased expression (<−0.5 log2 fold change) were compared with matrix stiffness regulated genes.

The online tool of the Gene Ontology Consortium (PANTHER Overrepresentation Test, release 20160715) was used to determine significantly enriched GO terms from a group of genes. The Bonferroni correction for multiple testing was applied and GO terms with $p < 0.05$ were included. For visualization of regulated GO terms and associated genes the Cytoscape software was employed. For visualization of Venn diagrams, the BioVenn web application was employed.

**Immunofluorescence.** For whole-mount immunostaining, tissues were fixed in 4% paraformaldehyde (PFA) for 2 h at RT or overnight at 4 °C, permeabilised in 0.3% Triton-X100 in PBS (PBSTx) and blocked in PBSTx plus 3% milk (blocking buffer). Primary antibodies were incubated at 4 °C overnight in blocking buffer. After washing in PBSTx, the samples were incubated with fluorocrome-conjugated secondary antibodies in blocking buffer, before further washing and mounting in Mowiol or Dako Fluorescence Mounting Medium. For visualization of cardinal veins and lymph sacs, 150 μm vibratome cross sections of E11.5 wild type or E12.5 *Tie2-Cre* embryos or 10 μm cryo-sections from E11 *Prox1-GFP* embryos were used for staining as described above. For staining of primary LECs, cells were fixed with 4% PFA in PBS for 20 min at RT and permeabilized using 0.5% Triton-X100 in PBS for 5 min at RT followed by blocking with 3% BSA in PBSTx for 1 h. Primary antibodies were incubated for 1 h at RT, washed twice with PBSTx and subsequently incubated with secondary antibodies for 45 min at RT before further washing and mounting in Dako Fluorescence Mounting Medium. The following antibodies were used: rabbit anti-GATA2 (cat. NBP1-82581, Novus Biologicals, 1:1000), rabbit anti-GATA2 (cat. sc-9008, Santa Cruz Biotechnology, 1:200), goat anti-VE-cadherin (C19, cat. sc-6458, Santa Cruz Biotechnology, 2 μg ml⁻¹), goat anti-LaminB (M20, cat. sc-6217, Santa Cruz Biotechnology, 1:300), rabbit anti-Collagen I (cat. ab34710, Abcam, 1 μg ml⁻¹), rat anti-LYVE1 (cat. MAB2125, R&D Systems, 1:100), rabbit anti-LYVE1 (cat. 103-PA50AG, RELIATech GmbH, 1:100), goat anti-Neuropilin2 (cat. AF567, R&D Systems, 1:200), hamster anti-Podoplanin (clone 8.1.1., Developmental Studies Hybridoma Bank, 1:1000), rabbit anti-PROX1[37] (1:200), goat anti-PROX1 (cat. AF2727, R&D Systems, 1:200), goat anti-VEGFR3 (cat. AF743, R&D Systems, 1:200), rat anti-Endomucin (cat. sc-65495, Santa Cruz Biotechnology, 1:500). Secondary antibodies conjugated to AF488, Cy2, Cy3 or Cy5 were obtained from Jackson ImmunoResearch (all used 1:200). Additionally, DAPI (D9542, Sigma Aldrich) and Alexa Fluor® 568 Phalloidin (A12380, ThermoFisher Scientific) were used. Whole-mount immunostaining for light sheet microscopy was performed as previously described[20]. In brief, Anti-PECAM1 (clone 5D2.6 and 1G5.2[79], 20 μg ml⁻¹), anti-VEGFR3 (cat. AF743, R&D Systems, 15 μg ml⁻¹) and anti-PROX1 (cat. 102-PA32, RELIATech GmbH, 1:200) were diluted in PermBlock. For visualization, embryo wholemount stainings were embedded in 1% low-melting point agarose. After dehydration in methanol (50, 70, 95, 100, 100% methanol, each step 30 min), samples were twice optically cleared in a benzyl alcohol:benzyl benzoate solution (BABB, ratio 1:2 (v/v)) for 4 h each.

**Image acquisition.** Confocal images of tissues and cells represent maximum intensity projections of Z-stacks that were acquired using Zeiss LSM 780 confocal microscope with Plan Apochromat DIC 10 × /0.45 NA or Plan Apochromat DIC 40 × /1.3 NA objectives and Zeiss Imager.Z2 and Zen 2012 software, or Leica SP8 inverted microscope with HCX PL APO CS 10 × /0.40 DRY, Fluotar VISIR 25 × / 0.95 water or HC PL APO CS2 63 × /1.30 GLYC objectives and Leica LAS-X software. Stereomicroscope images of embryos were acquired with Nikon SMZ1500 stereomicroscope equipped with a DS-5M Nikon camera.

BABB-cleared embryos were imaged using a LaVision UltraMicroscope II. Stacks were captured with a step size of 1 μm at different magnifications. 3D reconstruction and analysis of ultramicroscopy stacks were performed using the volume visualization framework Voreen (volume rendering engine)[73].

**Image quantification.** All quantifications were done using Fiji ImageJ unless differently stated. For Collagen I immunostaining pixel intensity measurements, corrected total cell fluorescence = integrated density − (area of selected cell × mean fluorescence of background readings) was automatically calculated (five images per embryo, 10–45 measurements per image from maximum intensity projection images of 36 μm z-stacks from two wild type E11 embryos). Nuclear circularity was measured from LaminB stained vibratome sections of two wild type E11 embryos. LaminB immunostaining of the nucleus was analyzed and compared between Endomucin positive ECs in the venous vessel wall to Endomucin negative cells outside the vessel wall (two embryos and two images per embryo, 7–10 measurements per single plane image in cells within and outside the vessel wall). For quantification of GATA2 staining intensity in transverse cryo sections of three E11 embryos, IMARIS software was used to create a 3D surface mask based on *Prox1-GFP* expression that was used to extract GATA2 and VEGFR3 pixel intensities of LECs within (4 images per embryo, 3–4 cells per image) and outside of the CV (5 images per embryo, 2–3 cells per image), from maximum intensity projection images of 50–100 z-stacks. For quantification of GATA2 staining intensity in human LECs grown on soft or stiff matrix, the GATA2 antibody (Novus Biologicals, NBP1-82581), that has previously been shown to specifically recognize GATA2, was used[18]. From 3 experiments cell coverage was determined by a defined threshold and pixel intensities were measured for $n = 8$ images with 8–24

cells per image (total 131 cells) on soft matrix, and for $n = 8$ images with 21–37 cells per image (total $n = 232$ cells) on stiff matrix, and the average pixel intensity value for each image was plotted. To distinguish cytoplasmic and nuclear pixel intensities, DAPI staining was used to generate a nuclear mask and subtracted from the overall pixel intensity of each image. For quantification of the number and length of LEC sprouts in the fibrin gel bead assay, 20–30 beads from two independent experiments were analyzed for each condition. Quantification of lymphatic vessel parameters (sprouting, diameter, branching, number of cells and filopodia) from whole-mount stained tissue was done using maximum intensity projection images of tile scanned E15.5 and E17.5 skin samples (xy = 3400 μm × 1700 μm, upper thoracic region). For measurement of the distance of lymphatic sprouts from the midline (i.e. open midline), dorsal midline was marked with a line and 5–8 measurements, 2–4 on each side of the midline, were taken from each image. Lymphatic vessels closest to the midline were always included. Vessel diameter was determined by measuring the thickest part of each vessel segment in between branch points. 22–47 measurements were taken from each skin samples. All branch points in each image were marked using Photoshop CS5 software and counted manually. Filopodia were quantified from Nrp2 stained lymphatic vessels tips (150 μm, $n = 3–4$ tips per embryo) in the sprouting front area. All filopodia were marked using Photoshop CS5 software and counted manually. Filopodia length was measured using Fiji ImageJ. Quantification of LEC numbers was done using CellProfiler Software (http://www.cellprofiler.org/). The pipeline included 1 major step: Identification of primary objects. LECs nuclei were identified and counted based on PROX1 staining. Resulting overlay images of the identified objects were saved and checked independently to verify accuracy.

**Ex vivo atomic force microscopy.** E11 *Prox1-GFP* embryos were chopped into 350 μm transverse slices with the McIllwain Tissue Chopper (Stoelting). Highest blade force and lowest chopping speed was used. Slices were carefully separated from each other with 27 gauge needles and transferred in PBS with a 3.5 ml transfer pipette (Sarstedt) and low suction to prevent tissue distortion. Tissue slices with relevant embryonic structures were transferred on Grade 105 Whatman Lens Cleaning Tissue (GE) floating on PBS. Whatman paper and the attached tissue slice were subsequently dried for 30 s on an absorbing tissue before it was transferred to a force spectroscopy suitable polypropylene culture dish that was covered with a thin layer of fluid 40 °C low melting agarose (6% in PBS). After 1 min the agarose solidified and the embryo tissue slice was fixed from underneath (through the Whatman paper) in the cell culture dish to allow tissue indentation measurements. Prior to force spectroscopy, the position of *Prox1-GFP* positive structures (cardinal vein and future primary lymphatic plexus area) and other features of interest were located using an upright fluorescence microscope (Nikon ME600) equipped with a mercury lamp and fluorescence camera (SPOT RT Monochrome, Diagnostic Instrument Inc, USA). After mounting in the atomic force microscope (JPK Nanowizard 3 with top-view optics, JPK Instruments, Berlin), force spectroscopy was performed after locating the measurement position via the top-view optics. To maximize the chance of landing on the correct area based on the fluorescence image, a matrix of 3 × 3 force spectroscopy points was set up over a 10 × 10 μm area. Account was taken of the cantilever angle (10°) and a form factor for v-shaped cantilevers. Calibration of the deflection sensitivity in liquid was performed by recording force ramps against a bare Petri dish in PBS buffer. The bare dish was then replaced with dishes containing the fixed embryo slices in PBS buffer for measurement of tissue stiffness via indentation ($n = 10$ measurements from one embryo (CV) and $n = 114$ (outside CV) measurements from three embryos). Five force ramps were performed at each position of a 3 × 3 point matrix over an area of 10 × 10 μm. A force setpoint of 5 nN and approach/retract speed of 2 μm s⁻¹ was used and the scan range was set to between 1 and 14 μm depending on the range of the forces measured. The Hertz indentation model was fitted to each approach curve to provide an effective Young's Modulus using the JPK Data Processing software. This was done by first correcting the measured curves for baseline position and slope, and then converting them to force versus separation curves. The model was fitted using square pyramidal indenter geometry, with an average edge angle of 25°. The values of Young's modulus determined via AFM should not be considered as quantitative absolute values because of the assumptions made in the Hertz model—namely that the indenter velocity is very low with respect to the indentation depth, and that the material is homogeneous and elastic (living cells are highly heterogeneous and dynamic structures). Force spectroscopy is a "quasi-static" technique, where the constant motion of the probe can give rise to viscous damping, which is difficult to accurately account for. However, as the same probe and indentation conditions were used for all measurements, the results obtained can be compared with each other to provide an effective Young's modulus where viscous damping may make some contribution to the measured value.

**Chromatin Immunoprecipitiation.** Human LECs were harvested and processed for ChIP using a truChIP™ Low Cell Chromatin Shearing Kit with SDS Shearing Buffer (Covaris). Briefly, 10 million cells per ml were cross-linked using 1% formaldehyde for 5 min, neutralized with glycine, lysed and nuclei washed. Sheared chromatin from 10 million cell equivalents was immunoprecipitated with 15 μg rabbit anti-GATA2 antibody (H-116 × , cat. sc-9008; Santa Cruz Biotechnology). Following washing and reversal of crosslinks, DNA was purified using a Qiagen MinElute PCR Purification Kit. ChIP-Seq library preparation and sequencing was carried out at the ACRF Cancer Genomics Facility, Center for Cancer Biology,

Adelaide. ChIP-Seq libraries were prepared according to Illumina's TruSeq Sample Prep Guide Revision A with some minor modifications. Briefly, DNA was end-repaired, followed by adenylation of the 3′ ends. Next, Illumina indexing adapters were ligated to the DNA. 18 cycles of PCR were performed to enrich for successfully adapter ligated molecules. Ligation products were purified with a 2% Pippin Prep gel (Sage Science), selecting a size range of 250–300 base pairs. The size distribution and yield of the purified libraries were determined using an Agilent Bioanalyzer High Sensitivity chip and Qubit dsDNA HS assay (Invitrogen), respectively. Libraries were pooled in equimolar ratios and sequenced in one lane of a HiSeq 2500 short read flowcell ($1 \times 50$ bp). Reads were mapped to the UCSC hg19 genome with BWA (71) version 0.7.9a-r786, allowing at most three alignments. Read depth was calculated via genomeCoverageBed from BEDTools, using the scale parameter to normalize to reads per million. Read depth was plotted relative to annotated transcription start sites via custom Python scripts utilizing Matplotlib.

**Statistics**. GraphPad Prism was used for graphic representation and statistical analysis of the data. We used 2-tailed unpaired Student's *t*-test to compare between two means, assuming equal variance; One-sample *t*-test to compare sample mean with a normalized control value = 1 (Figs. 2a, b, 3a, 5b, c, e); and Fisher's exact test to determine association between two categorical variables (Fig. 5g). Differences were considered statistically significant when $p < 0.05$. The experiments were not randomized. No blinding was done in the analysis and quantifications.

**Data availability**. All relevant data are available from the corresponding author upon request. RNA microarray data that support the findings of this study have been deposited in GEO (Gene Expression Omnibus) repository with the accession code GSE110726.

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

## Acknowledgements

We would like to thank Sagrario Ortega (CNIO, Madrid) for the *Vegfr3-CreER^T2* mice; Simon Stritt and Maria Ulvmar for help and advice with flow cytometry, respectively; Sofie Wagenius and Dimitrios Kokoretsis for technical assistance; Holger Gerhardt (Max Delbrück Center for Molecular Medicine, Berlin) for advice and discussions; and Ingvar Ferby (Ludwig Institute for Cancer Research, Uppsala) for critical comments on the manuscript. We also thank the Biovis facility at Uppsala University and FACS facility at the Francis Crick Institute for the help with flow cytometry experiments and the Array and Analysis Facility, at the Science for Life Laboratory at Uppsala Biomedical Center for performing the microarray experiments. This study was supported by the Swedish Research Council (D0368601 and 542-2014-3535), the Swedish Cancer Society (CAN 2013/387) and the European Research Council (ERC-2014-CoG-646849) (TM), the Francis Crick Institute which receives its core funding from Cancer Research UK (FC001057), the UK Medical Research Council (FC001057), and the Wellcome Trust (FC001057) (DPC), the British Heart Foundation (SP/13/5/30288; IMC, PO), the Australian Government National Health and Medical Research Council (APP1061365, NH), and DFG SFB656, SFB 629 and Cluster of Excellence EXC 1003 (FK). M.Fr. was supported by the postdoctoral fellowships from Lymphatic Education & Research Network and GA Johansson's Foundation.

## Author contributions

M.Fr., A.T., C.D., I.M.C., M.Fi., H.O. and J.K. performed experiments; M.Fr., A.T., C.D., I.M.C., M.Fi., H.O., J.K., L.H., N.H., F.K. and T.M. analyzed data; D.P.C. and P.O. and provided resources; M.S. provided essential tools; M.Fr. and T.M. wrote the paper; T.M. directed the study; all authors discussed the results and commented on the manuscript.

## Additional information

**Competing interests:** The authors declare no competing interests.

