## [Peer Review File · Nature Communications]

Reviewer #1:

Remarks to the Author:

The manuscript by Fry et al. describes so far little studied question, namely the role of matrix stiffness in the regulation of lymphatic endothelial cell (LECs) behavior. The authors analyze changes in gene expression between LECs cultured on stiff vs. soft matrices and identify the transcription factor Gata2 as being induced by the culture in soft matrix conditions. They further show that loss of Gata2 leads to defective lymphatic vascular development, using either constitutive deletion model (Tie2-Cre) or two inducible strains (Flt4-CreERT2 or Prox1-CreERT2). In the first case there is a rather severe defect of LEC sprouting from cardinal vein, but also defective blood vessel development. Inducible inactivation of Gata2 in LECs reduces sprouting and filopodia formations and results in more primitively organized, enlarged dermal lymphatic vascular network. The authors show that the expression of a key lymphangiogenic receptor Vegfr-3 is reduced upon Gata2 knockdown in vitro in soft matrix culture conditions and in ex vivo sorted dermal LECs. Further in vitro analyses show that loss of Gata2 impairs sprouting of LECs in response to Vegf-C.

This is an interesting work that provides new insights into the role of mechanotransduction in lymphangiogenesis. It also completes previous studies, which identified Gata2 as being induced by shear stress and important for lymphatic valve formation. Finally, it potentially explains why patients with Emberger syndrome have an early onset lymphatic vascular dysfunction.

Major points:

1. The title “ Matrix stiffness dictates ..” is, in my view, too general and does not reflect well the message of the paper. Very nice AFM data are provided however they correlate but do not prove that matrix stiffness is important for lymphangiogenic response in vivo. Furthermore, LEC autonomous inactivation of Gata2 reduces but does not abolishes lymphangiogenesis, whereas it is completely prevented in for example mice with inactivation of Vegf-c. The title is too general also because the same could be said of e.g. blood vessels or mammary gland formation. I would suggest modifying the title and including more specific information highlighting the role of Gata2; this is important because the authors report that blood endothelial cells and LECs respond differently to the same mechanical cues.
2. The main hypothesis of the manuscript is that the induction of Gata2 by low stiffness is necessary to maintain Vegfr-3 expression in LECs. This is clearly demonstrated in vitro, however the manuscript would be stronger if the authors could provide some additional evidence from in vivo analyses – for example, does Gata2 expression increases when LECs emigrate from cardinal vein? Do these cells also have more cytoplasmic Gata2?
3. There is a somewhat of a discrepancy between changes in Gata2 mRNA levels and nuclear Gata2 accumulation in different stiffness conditions: clearly, mRNA is induced in soft matrix but there is almost no change in nuclear Gata2 levels (may be 20% increase) – in contrast cytoplasmic Gata2 levels are much higher. The authors should consider alternative hypotheses, such as regulation of not only expression but also transcriptional activity, e.g. via sumoylation, known to regulate Gata2.

4. Clearly, the motility of LECs is affected both in vitro and in vivo; it will be important to complete these data by the analyses of the effects of Gata2 loss of LEC proliferation and survival, especially in vivo.
5. Although GATA2 Chip-seq is not necessarily important at this stage for conveying the main message of the manuscript, it would be informative to analyze the promoters of the genes regulated by stiffness (and Gata2) to further to identify which transcriptional modules are enriched in up- and down-regulated genes.
6. Is less than 50% down regulation of Vegfr-3 in LECs is sufficient to fully explain the phenotype? Vegfr-3 heterozygous mice have normal lymphatics, undoubtedly, additional Gata2 target genes are also contributing. It will be important to verify whether FGF receptor signaling recently shown to be important for embryonic lymphangiogenesis, is also reduced in Gata2 deficient LECs.
7. Is the phenotype of Gata2 inactivation limited to skin?

Technical points:

1. Supplementary Table 1. Matrix stiffness regulated transcription factors – please provide the LFC and statistics for the transcription factors in the table.
2. According to the Nature journal policy gene expression profiling results should be deposited to Gene Expression Omnibus (GEO).
3. Comparison of common and unique mechanoresponses (Fig.7a) is based on in vitro experiments; while interesting for demonstrating differential responses of endothelial cells to various mechanical stimuli, I believe that it does not provide enough data for scheme in 7b – the latter is interesting, but speculative, and in my view, is better suited for a review or opinion article.

Reviewer #2:

Remarks to the Author:

This manuscript reveals a novel mechanism by which matrix stiffness regulates lymphatic morphogenesis. They show that GATA2 is activated by soft matrix and that this activation is required for early lymphatic vascular morphogenesis via regulation of VEGFR3 and VEGF-C.

-Stiffness has been shown to regulate levels of LaminA, the authors should quantitate possible differences in expression levels of LaminA between cells of the CV and surrounding tissue.

- are proliferation and YAP/TAZ gene targets GATA2 -dependent?

- the authors measured stiffness by AFM at E11 and showed differences in the CV vs the surrounding tissue. How does stiffness compare later on in development when lymph flow has started?

-the finding that low stiffness- or OSS-induced upregulation of GATA2 results in initiation of valve morphogenesis but by distinct transcriptional mechanisms is very interesting and worthy of further investigation. This is particularly relevant considering the opposing effects of low stiffness and OSS on the YAP/TAZ pathway. How do the authors explain that two (almost opposite) mechanical stimuli result in up regulation of GATA2 and similar initiation of valve morphogenesis via such different transcriptional effects?

- how does GATA2 regulate VEGFR3 expression?

We thank the Reviewers for their constructive criticism that helped us to improve the manuscript. We have addressed all Reviewers' concerns point by point as explained below, and added new experimental data to strengthen our conclusions. In particular, our new data show that:

- GATA2 mRNA and protein expression is increased *in vivo* in LECs that have migrated out of the cardinal vein.
- GATA2 regulates VEGFR3 expression in LECs by directly binding to a site in intron 1 of *FLT4* (encoding VEGFR3) that co-localizes with H3K27Ac and a DNase hypersensitivity region that indicate enhancer activity.
- Matrix stiffness induced regulation of proliferation and YAP/TAZ target genes is not controlled by GATA2. Together with the data presented in the original manuscript, this suggests an important role of GATA2 in regulating soft matrix induced transcriptional response controlling migration, but not the general mechanoreponse in LECs.
- *Gata2* deficient LECs show down-regulation of other previously described lymphatic regulators, *Fgfr3* and *Efnb2*¹⁻³. Interestingly, like GATA2, both genes were also up-regulated by soft matrix. Besides VEGFR3, these genes may thus contribute to the lymphatic vascular defects in the *Gata2* deficient mice.

Specific points were addressed as follows (in blue, new figure panels indicated in bold):

Reviewer #1 (Remarks to the Author)

The manuscript by Fry et al. describes so far little studied question, namely the role of matrix stiffness in the regulation of lymphatic endothelial cell (LECs) behavior. The authors analyze changes in gene expression between LECs cultured on stiff vs. soft matrices and identify the transcription factor Gata2 as being induced by the culture in soft matrix conditions. They further show that loss of Gata2 leads to defective lymphatic vascular development, using either constitutive deletion model (Tie2-Cre) or two inducible strains (Flt4-CreERT2 or Prox1-CreERT2). In the first case, there is a rather severe defect of LEC sprouting from cardinal vein, but also defective blood vessel development. Inducible inactivation of Gata2 in LECs reduces sprouting and filopodia formations and results in more primitively organized, enlarged dermal lymphatic vascular network. The authors show that the expression of a key lymphangiogenic receptor Vegfr-3 is reduced upon Gata2 knockdown in vitro in soft matrix culture conditions and in ex vivo sorted dermal LECs. Further in vitro analyses show that loss of Gata2 impairs sprouting of LECs in response to Vegf-C.

This is an interesting work that provides new insights into the role of mechanotransduction in lymphangiogenesis. It also completes previous studies, which identified Gata2 as being induced by shear stress and important for lymphatic valve formation. Finally, it potentially explains why patients with Emberger syndrome have an early onset lymphatic vascular dysfunction.

Major points:

1. The title “Matrix stiffness dictates ..” is, in my view, too general and does not reflect well the message of the paper. Very nice AFM data are provided however they correlate but do not prove that matrix stiffness is important for lymphangiogenic response *in vivo*. Furthermore, LEC autonomous inactivation of *Gata2* reduces but does not abolish lymphangiogenesis, whereas it is completely prevented in for example mice with inactivation of *Vegf-c*. The title is too general also because the same could be said of e.g. blood vessels or mammary gland formation. I would suggest modifying the title and including more specific information highlighting the role of *Gata2*; this is important because the authors report that blood endothelial cells and LECs respond differently to the same mechanical cues.

Response: We have followed the Reviewer’s advice and changed the title to highlight the role of GATA2. The new title “Matrix stiffness-induced GATA2 regulation controls lymphangiogenic growth factor responsiveness and lymphatic vessel formation” specifically emphasizes the role of GATA2 in LECs.

2. The main hypothesis of the manuscript is that the induction of *Gata2* by low stiffness is necessary to maintain *Vegfr-3* expression in LECs. This is clearly demonstrated *in vitro*, however the manuscript would be stronger if the authors could provide some additional evidence from *in vivo* analyses – for example, does *Gata2* expression increase when LECs emigrate from cardinal vein? Do these cells also have more cytoplasmic *Gata2*?

Response: Following the Reviewer’s suggestion, we analyzed GATA2 expression *in vivo* both at the mRNA and protein level and provide new data showing that GATA2 is upregulated in LECs that have migrated out of the CV.

First, we sorted PECAM1⁺CD45⁻ ECs from E11 embryos by flow cytometry and analyzed mRNA expression in ECs of the CV and forming lymphatic vessels. To this end, we dissected jugular regions of E11 embryos carrying the *Prox1-GFP* transgene that labels ECs within the CV, including LEC progenitors, as well as LECs outside of the CV (Fig. 1a). The latter were further identified by expression of PDPN⁺. mRNA expression in sorted ECs was normalized to the pan-endothelial marker *Tek*, which was not regulated in LECs by matrix stiffness (Supplementary Fig. 1c). As expected, *Prox1-GFP*⁺PDPN⁺ ECs showed a strong increase in the expression of the LEC markers *Vegfr3* and *Pdpn* when compared to their expression in *Prox1-GFP*⁺PDPN⁻ ECs (Fig. 3a). *Gata2* levels were also increased in the PDPN⁺ LECs by 1.7-fold (Fig. 3a). Expression of the pan-endothelial marker *Erg* was not changed (Fig. 3a).

GATA2 protein expression was further assessed by immunostaining of transverse cryo sections of E11 embryos. Staining intensity was quantified after masking the *Prox1-GFP* signal to extract the endothelial GATA2 and VEGFR3 signal. As previously reported^{4,5}, VEGFR3 was highly expressed in LECs outside of the CV, showing a 2-fold increase in signal intensity compared to ECs within the CV (Fig. 3b). GATA2 immunostaining intensity was 1.9-fold higher in *Prox1-GFP*⁺VEGFR3^{high} LECs outside of the CV compared to *Prox1-GFP*⁺VEGFR3^{low} ECs within the CV (Fig. 3b). Due to low expression levels and expression in non-LECs, it was not possible to confidently distinguish GATA2 expression in the nucleus from the expression in cytoplasm.

3. There is a somewhat of a discrepancy between changes in *Gata2* mRNA levels and nuclear *Gata2* accumulation in different stiffness conditions: clearly, mRNA is induced in soft matrix

but there is almost no change in nuclear *Gata2* levels (may be 20% increase) – in contrast cytoplasmic *Gata2* levels are much higher. The authors should consider alternative hypotheses, such as regulation of not only expression but also transcriptional activity, e.g. via sumoylation, known to regulate *Gata2*.

Response: We repeated the GATA2 immunostainings with an antibody that has been reported to specifically detect GATA2 protein *in vivo* in tissue sections⁶. We have further verified the specificity of this new antibody using GATA2 siRNA (data not shown) as we suspect that the GATA2 antibody we used previously may also detect GATA3. With the new antibody, GATA2 expression was found to be 1.8-fold higher in the nucleus and 2.1-fold in the cytoplasm in human LECs seeded on soft in comparison to stiff matrix (**Fig. 2c, d**). A similar, approximately 2-fold upregulation of GATA2 expression was previously reported in human LECs in response to oscillatory flow and shown to be functionally important⁶.

We also followed the Reviewer's suggestion and analyzed the sumoylation state of GATA2 in human LECs seeded on stiff and soft matrices. GATA2 sumoylation was assessed as recently described⁷. We were not able to detect sumoylated GATA2 by either immunoprecipitating GATA2 (**Figure 1 for Reviewers**) or SUMO-1 (data not shown) from cell lysates. We conclude that sumoylation may not be involved in controlling matrix stiffness regulated GATA2 expression in the LECs.

Figure 1 for the Reviewers. GATA2 sumoylation is not detectable in human LECs. Human LECs were seeded on stiff (25 kPa) or soft (0.2 kPa) matrix and sumoylation was analysed by immunoprecipitation of GATA2, in the presence of 20mM N-ethylmaleimide (deSUMOylase inhibitor). Subsequently, SUMO-1 immunoblotting was performed. Sumoylated GATA2 was not detectable (expected molecular weight approximately 100 kDa⁷). Whole cell lysates (WCL) were blotted for α -tubulin and VE-cadherin to verify equal loading.

4. Clearly, the motility of LECs is affected both *in vitro* and *in vivo*; it will be important to complete these data by the analyses of the effects of *Gata2* loss of LEC proliferation and survival, especially *in vivo*.

Response: We showed that the total LEC number was not altered in the skin of *Gata2* mutants (Fig. 6j), suggesting that LEC proliferation or survival was not affected. To strengthen this data, we have analyzed the expression of proliferation markers *Mki67* and *Ccnb1* in freshly sorted dermal LECs from E15.5 *Gata2* mutant and littermate control embryos and did not observe differences between the genotypes (Fig. 6k).

5. Although GATA2 Chip-seq is not necessarily important at this stage for conveying the main message of the manuscript, it would be informative to analyze the promoters of the genes regulated by stiffness (and *Gata2*) to further to identify which transcriptional modules are enriched in up- and down-regulated genes.

Response: To address this question, we employed the TFactS online module (<https://omictools.com/tfacts-tool>) to predict which are “the transcription factors that are significantly regulated in a biological condition based on lists of up-regulated and down-regulated genes resulted from transcriptomics experiments”⁸. We have highlighted the genes, which include GATA2, as core transcription factors controlling LEC response to matrix stiffness in the revised **Supplementary Table 1**, which is now supplied as an Excel file.

6. Is less than 50% down regulation of *Vegfr-3* in LECs is sufficient to fully explain the phenotype? *Vegfr-3* heterozygous mice have normal lymphatics, undoubtedly, additional *Gata2* target genes are also contributing. It will be important to verify whether FGF receptor signaling recently shown to be important for embryonic lymphangiogenesis, is also reduced in *Gata2* deficient LECs.

Response: The expression and activity of the VEGF-C - VEGFR3 pathway has to be tightly controlled to ensure normal lymphatic development. This is highlighted by the finding that loss of a single allele of *Vegfc* leads to lymphatic hypoplasia in mice⁵, and heterozygous inactivating mutations in *FLT4* (encoding VEGFR3) are causative of primary lymphedema (Milroy disease) in human⁹ and in mice¹⁰. Mutations in genes in the VEGFR3 pathway explain altogether 36% of familial lymphedema¹¹. We show that GATA2 knock down, leading to a 27% decrease in the baseline expression levels of VEGFR3 (Fig. 5e) and a complete inhibition of the soft matrix-induced VEGFR3 upregulation (Fig. 5c), is sufficient to block LEC response to VEGFC in an *in vitro* sprouting assay (Fig. 5f). Although VEGFR3 regulation by GATA2 is thus arguably of critical importance in regulating LEC response to the major lymphangiogenic growth factor VEGF-C, the Reviewer is right in that a number of other GATA2 targets may also contribute, similar to the role of GATA2 in controlling a set of genes required for the formation of lymphatic valves^{6,12}.

Following the Reviewer’s suggestion, we analyzed the potential involvement of FGF receptor signaling in the *Gata2* mutant phenotype. To this end, we sorted dermal LECs by FACS from E15.5 *Gata2* mutant and littermate control embryos, and analyzed mRNA expression of the four FGF receptors. *Fgfr3* mRNA levels were down-regulated in LECs isolated from *Gata2* deficient in comparison to control LECs, whereas *Fgfr1* and *Fgfr2* were up-regulated (Fig. 5a, Figure 2 for the Reviewers). *Fgfr4* was not expressed in dermal LECs at E15.5 (data not shown). FGFR3 was previously shown to co-operate with FGFR1 to regulate early lymphatic development². Upregulation *Fgfr1* and *Fgfr2* in *Gata2* deficient LECs may thus compensate

for the reduction in *Fgfr3* levels. Interestingly, analysis of RNA array data showed that like *GATA2*, *FGFR3* expression is increased in LECs grown on soft matrix (**Fig. 4b**). We now highlight the *GATA2* and soft matrix dependent regulation of *FGFR3* both in the discussion and the revised summary figure (**Fig. 7b**).

Figure 2 for the Reviewers. FGFR regulation in *Gata2* deficient LECs *in vivo*. Relative mRNA expression levels of *Fgfr1*, *Fgfr2* and *Fgfr3* in freshly isolated LECs from E15.5 control and *Gata2* mutant (n=3-4 embryos from two different litters). Horizontal lines represent mean \pm s.e.m. p value, unpaired Student's t-test.

7. Is the phenotype of *Gata2* inactivation limited to skin?

Response: We show that *GATA2* is required for the development of two lymphatic vessel networks that form at different embryonic stages through a process of LEC migration and/or vessel sprouting; the first primitive lymphatic vessels in the jugular region of the embryo (Fig. 3) and the lymphatic vasculature of the dorsal skin (Fig. 6). In the mesentery, where lymphatic vessels form by assembly from progenitors (lymphvasculogenesis)¹³, initial vessel formation is not affected but valve morphogenesis is blocked (⁶; data not shown). It is therefore possible that loss of *Gata2* mainly affects tissues where lymphatic vasculature forms by sprouting. We now discuss this on page 18-19.

Technical points:

1. Supplementary Table 1. Matrix stiffness regulated transcription factors – please provide the LFC and statistics for the transcription factors in the table.

Response: Following the Reviewer's suggestion, we have included additional information in Supplementary Table 1, which is now provided as an excel file; 1) average fold change/log2 fold change/average expression (average log2 intensity), which are given separately for each exon set ID that passed the criteria for inclusion, 2) t-test P-value from pairwise comparison of the 6 replicates, and 3) original raw log2-intensity data for all 6 replicates.

We did not use the P-value to define regulated genes. As described in the methods and briefly below, an alternative approach was used to summarize the differential expression at gene-level from the exon-array datasets:

‘To identify differentially expressed genes between the ctrl stiff and ctrl soft groups, a stepwise analysis with 6 biological replicates was performed. First, exon set ID’s with an average expression lower than 5 were considered as not significantly expressed and excluded from the analysis. A threshold of 40% increase ($>0.5 \log_2$ fold change) or decrease ($<-0.5 \log_2$ fold change) of gene expression on the soft matrix (vs. stiff matrix) was considered for further analysis. For all genes with 3 or more exon probe set ID’s regulated above the defined thresholds, the average \log_2 fold change of the regulated exon probe ID’s was calculated and used to generate the final list of genes regulated by matrix stiffness. 3.8% of the regulated genes were found to be both increased and decreased, potentially indicating differential expression of different splice variants. These genes were excluded from further validations. To identify regulated transcription factors the most updated database of transcription factors was exported. To determine which of the genes are regulated in a GATA2 dependent manner, a stepwise analysis with 3 biological replicates ctrl siRNA soft vs GATA2 siRNA soft groups was performed. The previously identified exon probe ID’s for genes differentially regulated between the ctrl stiff and ctrl soft groups were extracted and their average \log_2 fold change was calculated for the new data set. A threshold of 40% increase ($>0.5 \log_2$ fold change) or decrease ($<-0.5 \log_2$ fold change) of gene expression in the absence of the GATA2 was used to generate the final list of genes.’

2. *According to the Nature journal policy gene expression profiling results should be deposited to Gene Expression Omnibus (GEO).*

Response: We will deposit the data, including the original raw data files, to GEO when the manuscript has been accepted for publication.

3. *Comparison of common and unique mechano-responses (Fig.7a) is based on in vitro experiments; while interesting for demonstrating differential responses of endothelial cells to various mechanical stimuli, I believe that it does not provide enough data for scheme in 7b – the latter is interesting, but speculative, and in my view, is better suited for a review or opinion article.*

Response: We agree with the Reviewer and now summarize only our own findings in Fig. 7b.

Reviewer #2 (Remarks to the Author):

This manuscript reveals a novel mechanism by which matrix stiffness regulates lymphatic morphogenesis. They show that GATA2 is activated by soft matrix and that this activation is required for early lymphatic vascular morphogenesis via regulation of VEGFR3 and VEGF-C.

-Stiffness has been shown to regulate levels of LaminA, the authors should quantitate possible differences in expression levels of LaminA between cells of the CV and surrounding tissue.

Response: Since LaminB is the major embryonic Lamin¹⁴, we used LaminB staining to

determine nuclear shape, which was used as supporting data to show differences in the stiffness of the CV and the surrounding tissue. Following the Reviewer's suggestion, we also analyzed the expression of LaminA. However, we were not able to detect LaminA expression in E11 embryos (data not shown). This is consistent with previous studies showing that in mouse embryos, Lamin A/C expression first appears at E12¹⁵.

- are proliferation and YAP/TAZ gene targets GATA2 –dependent?

Response: To answer this question, we repeated the experiment in Fig. 4a and analyzed the impact of *GATA2* knock-down on the expression of four proliferation markers and three direct YAP/TAZ target genes. Proliferation markers (*CCNB2*, *MIK67*, *MYBL2*, *BUB1*) and YAP/TAZ target genes (*CTGF*, *CYR61*) were not regulated by *GATA2* (**Supplementary Fig. 3b**). In addition, although reduced baseline expression of YAP/TAZ target *ANKRD1* in *GATA2* siRNA treated samples suggests its general regulation by *GATA2*, soft matrix-induced downregulation still occurred (**Supplementary Fig. 3b**). These results demonstrate that matrix stiffness induced regulation of proliferation and YAP/TAZ target genes in LECs is not controlled by *GATA2*.

- the authors measured stiffness by AFM at E11 and showed differences in the CV vs the surrounding tissue. How does stiffness compare later on in development when lymph flow has started?

Response: Tissue stiffness increases during embryonic development and is linked to gradual organization and enrichment of collagen fibers¹⁶⁻¹⁸. In addition to general stiffening of the tissue, local changes in matrix stiffness are likely of critical importance. It is interesting to note that soft, but not stiff matrix has been shown to precisely regulate cellular functions such as adhesion and invasiveness in a scale range of 0.1 kPa¹⁹, suggesting that small changes in stiffness in soft (e.g. embryonic) tissues can have profound effects on cellular behaviors. In the vasculature, the deposition and assembly of the basement membrane on basal endothelial surfaces is expected to dramatically change the mechanical properties of their substrate. It is tempting to speculate that migrating tip cells leading the vascular sprouts experience a softer matrix when not supported by the basement membrane of the vessel wall. Notably, *VEGFR3* is highly expressed in and indispensable for the function of tip cells^(20; and unpublished data). As it is difficult to experimentally prove this hypothesis we do not have evidence for stiffness-regulated expression of *GATA2* and *VEGFR3* in lymphatic vessel sprouts. However, it should be noted that *in vitro* deletion of *Gata2* in mouse LECs grown on stiff matrix showed a 27% reduction in *Vegfr3* expression (Fig. 5e). The ability of *GATA2* to regulate baseline levels of *VEGFR3* expression may thus explain dermal lymphatic vessel sprouting defects in the *Gata2* deficient embryos. We have clarified this in the revised manuscript (page 21).

-the finding that low stiffness- or OSS-induced upregulation of GATA2 results in initiation of valve morphogenesis but by distinct transcriptional mechanisms is very interesting and worthy of further investigation. This is particularly relevant considering the opposing effects of low stiffness and OSS on the YAP/TAZ pathway. How do the authors explain that two (almost opposite) mechanical stimuli result in up regulation of GATA2 and similar initiation of valve morphogenesis via such different transcriptional effects?

Response: We agree with the Reviewer that the observation that *GATA2* is regulated by and controls cellular responses to two different mechanical stimuli with opposing effects on the

YAP/TAZ pathway is very interesting. We feel that the mechanism of GATA2 regulation by the different mechanical stimuli is however out of the scope of this study that focuses on matrix stiffness regulation of lymphatic development.

It has been described in several cell types including ECs that GATA2 interacts with other transcriptional regulators, including ETV2²¹ and LMO2²², to form multimeric transcription complexes. An interesting possibility that should be addressed in future studies is whether OSS- and soft matrix-induced differences in the GATA2 mediated regulation of target genes and cellular responses is explained by formation of different transcriptional complexes. We have included discussion on this in the revised manuscript (page 19).

- how does GATA2 regulate VEGFR3 expression?

Response: To address this question, we collaborated with Natasha Harvey's group who performed GATA2 ChIPseq in human dermal LECs. One pronounced peak within a region covering the *FLT4* gene (encoding VEGFR3) and up to 50kb upstream of the *FLT4* promoter was identified (**Fig. 5d**). The GATA2 binding peak mapped to intron 1 of *FLT4* and co-localized with two indicators of active enhancer elements, H3K27Ac peak and a DNase hypersensitivity site (**Supplementary Fig. 5**). These data suggest that GATA2 regulates VEGFR3 expression in LECs by directly binding to a site in intron 1 of *FLT4*.

References:

1. Shin, J. W. *et al.* Prox1 promotes lineage-specific expression of fibroblast growth factor (FGF) receptor-3 in lymphatic endothelium: a role for FGF signaling in lymphangiogenesis. *Mol. Biol. Cell* **17**, 576–584 (2006).
2. Yu, P. *et al.* FGF-dependent metabolic control of vascular development. *Nature* **545**, 224–228 (2017).
3. Wang, Y. *et al.* Ephrin-B2 controls VEGF-induced angiogenesis and lymphangiogenesis. *Nature* **465**, 483–486 (2010).
4. Hägerling, R. *et al.* A novel multistep mechanism for initial lymphangiogenesis in mouse embryos based on ultramicroscopy. *EMBO J.* **32**, 629–644 (2013).
5. Karkkainen, M. J. *et al.* Vascular endothelial growth factor C is required for sprouting of the first lymphatic vessels from embryonic veins. *Nat. Immunol.* **5**, 74–80 (2004).
6. Kazenwadel, J. *et al.* GATA2 is required for lymphatic vessel valve development and maintenance. *J. Clin. Invest.* **125**, 2979–2994 (2015).
7. Chun, T.-H., Itoh, H., Subramanian, L., Iñiguez-Lluhí, J. A. & Nakao, K. Modification of GATA-2 transcriptional activity in endothelial cells by the SUMO E3 ligase PIASy. *Circ. Res.* **92**, 1201–1208 (2003).
8. Essaghir, A. *et al.* Transcription factor regulation can be accurately predicted from the presence of target gene signatures in microarray gene expression data. *Nucleic Acids Res.* **38**, e120 (2010).
9. Karkkainen, M. J. *et al.* Missense mutations interfere with VEGFR-3 signalling in primary lymphoedema. *Nat. Genet.* **25**, 153–159 (2000).
10. Karkkainen, M. J. *et al.* A model for gene therapy of human hereditary lymphedema. *Proc. Natl. Acad. Sci. U. S. A.* **98**, 12677–12682 (2001).
11. Mendola, A. *et al.* Mutations in the VEGFR3 signaling pathway explain 36% of familial lymphedema. *Mol. Syndromol.* **4**, 257–266 (2013).

12. Kazenwadel, J. *et al.* Loss-of-function germline GATA2 mutations in patients with MDS/AML or MonoMAC syndrome and primary lymphedema reveal a key role for GATA2 in the lymphatic vasculature. *Blood* **119**, 1283–1291 (2012).
13. Stanczuk, L. *et al.* cKit Lineage Hemogenic Endothelium-Derived Cells Contribute to Mesenteric Lymphatic Vessels. *Cell Rep.* (2015). doi:10.1016/j.celrep.2015.02.026
14. Osmanagic-Myers, S., Dechat, T. & Foisner, R. Lamins at the crossroads of mechanosignaling. *Genes Dev.* **29**, 225–237 (2015).
15. Röber, R. A., Weber, K. & Osborn, M. Differential timing of nuclear lamin A/C expression in the various organs of the mouse embryo and the young animal: a developmental study. *Dev. Camb. Engl.* **105**, 365–378 (1989).
16. Majkut, S. *et al.* Heart-specific stiffening in early embryos parallels matrix and myosin expression to optimize beating. *Curr. Biol. CB* **23**, 2434–2439 (2013).
17. Iwashita, M., Kataoka, N., Toida, K. & Kosodo, Y. Systematic profiling of spatiotemporal tissue and cellular stiffness in the developing brain. *Dev. Camb. Engl.* **141**, 3793–3798 (2014).
18. Chevalier, N. R. *et al.* How Tissue Mechanical Properties Affect Enteric Neural Crest Cell Migration. *Sci. Rep.* **6**, 20927 (2016).
19. Gu, Z. 0.1 kilopascal difference for mechanophenotyping: soft matrix precisely regulates cellular architecture for invasion. *Bioarchitecture* **4**, 116–118 (2014).
20. Tammela, T. *et al.* Blocking VEGFR-3 suppresses angiogenic sprouting and vascular network formation. *Nature* **454**, 656–660 (2008).
21. Shi, X. *et al.* Cooperative interaction of Etv2 and Gata2 regulates the development of endothelial and hematopoietic lineages. *Dev. Biol.* **389**, 208–218 (2014).
22. Coma, S. *et al.* GATA2 and Lmo2 control angiogenesis and lymphangiogenesis via direct transcriptional regulation of neuropilin-2. *Angiogenesis* **16**, 939–952 (2013).

Reviewer #1:

Remarks to the Author:

Minor comments:

1. Fig 2d - please indicate the values of individual samples as shown in Fig.2a
2. Measurements of Young's Modulus uses $n=10$ and $n=114$. The n probably corresponds to the number of individual AFM measurements and not number of embryos? How many individual embryos were analysed in total for AFM experiments?

Reviewer #2:

Remarks to the Author:

The authors have satisfactorily addressed my comments. This is a nice piece of work that identifies a novel mechanism by which matrix stiffness controls lymphatic vascular morphogenesis.

Response to the reviewers' comments

Reviewer #1 (Remarks to the Author):

Minor comments:

1. Fig 2d - please indicate the values of individual samples as shown in Fig.2a

We have added the individual data points, and information in the methods section (p. 32) and Fig. 2d legend: from 3 experiments pixel intensities were measured for n=8 images with 8-24 cells per image (total 131 cells) on soft matrix, and for n=8 images with 21-37 cells per image (total n=232 cells) on stiff matrix, and the average pixel intensity value for each image was plotted.

2. Measurements of Young's Modulus uses n=10 and n=114. The n probably corresponds to the number of individual AFM measurements and not number of embryos? How many individual embryos were analysed in total for AFM experiments?

This information was indeed missing and has now been added both in the methods section (p. 33) and Fig. 1d legend: n=10 measurements from 1 embryo (CV) and n=114 (outside CV) measurements from 3 embryos.

Reviewer #2 (Remarks to the Author):

The authors have satisfactorily addressed my comments. This is a nice piece of work that identifies a novel mechanism by which matrix stiffness controls lymphatic vascular morphogenesis.